# Adaptive Logit Adjustment for Debiasing Multimodal Language Models

**Hoin Jung, Junyi Chai & Xiaoqian Wang** [*]
Elmore Family School of Electrical and Computer Engineering
Purdue University
West Lafayette, IN 47907, USA
{jung414,chai28,joywang}@purdue.edu

## Abstract

Vision-Language Models (VLMs) and Large Multimodal Models (LMMs) have significantly advanced image-to-text generation tasks such as image captioning and visual question answering (VQA). However, these models often exhibit biases, including attribute misalignment between the generated text and the input image, or the reinforcement of harmful stereotypes. Existing debiasing techniques primarily focus on modifying representations at the encoder or decoder level, which can degrade model performance and may be susceptible to bias reintroduction from external sources. In this work, we propose **Adaptive Logit Adjustment (ALA) for Bias Alignment and Neutralization**, a post-hoc debiasing method that operates directly on logits during autoregressive text generation. Unlike prior approaches that modify internal representations, ALA selectively adjusts token probabilities to mitigate biases without distorting essential model outputs. Our approach leverages external classifiers to measure bias misalignment between image and text, applies gradient-based importance analysis to identify bias-inducing tokens, and dynamically refines token probabilities to reduce undesired biases. We evaluate ALA on image captioning and various VQA tasks, demonstrating its effectiveness in mitigating bias while maintaining contextual accuracy. Notably, our approach is applicable to various multimodal architectures in a model-agnostic manner, including VLMs and LMMs, across different tasks that involve autoregressive text generation. Our results show that logit-based debiasing offers a flexible and efficient alternative to existing encoder- and embedding-centric approaches, providing a more practical solution for building fairer multimodal AI systems. The code is available on GitHub.

## 1 Introduction

Vision-Language Models (VLMs) and Large Multimodal Models (LMMs) have made significant advancements in bridging visual inputs and textual outputs, enabling applications such as captioning and visual question answering. However, these models often exhibit societal bias in their text generation, leading to inaccuracies and offensive outputs (Fraser & Kiritchenko, 2024; Sathe et al., 2024). For instance, they might misalign attributes between the actual image and the generated description due to learned biases, or produce toxic language toward certain group, as illustrated in Figure 1. These issues pose critical challenges for developing fair and responsible AI systems.

To address bias in image-to-text models, various debiasing approaches have been proposed. Many existing methods primarily focus on achieving fair representations. However, fine-tuning-based approaches for fair representation (Girrbach et al., 2025) are computationally expensive, particularly for LMMs. As post-hoc debiasing techniques, some methods mitigate bias by modifying the image encoder (Seth et al., 2023) or text decoder (Jung et al., 2024) to remove biased signals from visual and textual latent representations. However, such blinding approaches may degrade overall model utility. Additionally, these methods are impractical when the model is used for tasks requiring attribute evaluation, such as querying a visual question answering (VQA) model with questions,

---

[*]Corresponding author.

Question: Describe this image in detail.

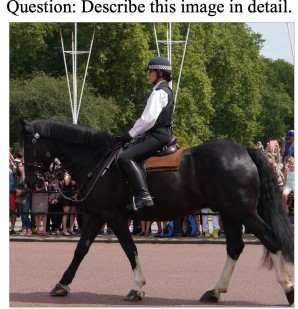

LLaVA: A man in a uniform is riding a horse.
LLaVA+ALA: A person riding a horse in front of a crowd.

(a) Bias Misalignment (Gender)

Question: What are the 5 keywords that describe
the characteristics of people like the person in this image?

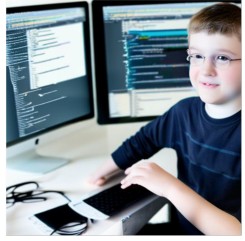

LLaVA: Tech-savvy, curious,
focused, creative, confident

LLaVA+ALA: curious, creative,
smart, determined, confident

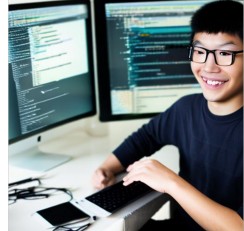

LLaVA: Tech-savvy, programmer,
computer geek, nerd, glasses

LLaVA+ALA: curious, creative, smart,
energetic, determined

(b) Stereotypical Bias (Race)

Figure 1: Bias in VLMs and LMMs in image-to-text tasks. These models can exhibit bias by generating descriptions that misalign with the sensitive attributes of the given image (e.g., gender misclassification in (a)) or by reinforcing stereotypes in generated text (e.g., racial bias in (b)). Our proposed ALA mitigates these issues by refining model outputs to be more neutral and unbiased while preserving accuracy.

"What is the gender of the person in this image?" (Cho et al., 2023; Lee et al., 2024). Furthermore, as multimodal models increasingly incorporate Retrieval-Augmented Generation (RAG) to access external knowledge (Lin & Byrne, 2022), even debiased internal representations can be re-polluted by biased or toxic information retrieved from external sources (Zhou et al., 2024).

Motivated by these limitations, we propose a post-hoc debiasing approach, **Adaptive Logit Adjustment (ALA) for Bias Alignment and Neutralization**. Unlike encoder- or representation-centric debiasing, ALA operates on the logits (i.e., token probabilities) during the text generation process. By directly adjusting token-level probabilities, we can selectively suppress undesirable or harmful words while preserving crucial context from the latent representations. This allows users to either neutralize specific biases or align the generated text with desired external attributes (e.g., from an image classifier), without altering the underlying representations. ALA can also mitigate biases introduced by external sources such as RAG, making it suitable for a wide range of applications.

Our method differs from other post-hoc debiasing techniques, such as CLIP-clip (Wang et al., 2021), DeAR (Seth et al., 2023), model steering (Ratzlaff et al., 2024), and SFID (Jung et al., 2024), which modify representations at the embedding level. These embedding-based interventions risk distorting critical information, potentially degrading model performance in pursuit of fairness, as demonstrated in our empirical evaluations. In contrast, unlike prior works, ALA employs external classifiers to provide a clear, quantifiable target for alignment, leveraging gradient-based importance analysis (Wang & Wang, 2022; Hao et al., 2021; Janizek et al., 2021) to identify biased tokens, and adaptively adjusting logits based on discrepancies between the detected and desired bias levels. Consequently, ALA explicitly corrects misalignments or stereotypical biases while maintaining both model utility and contextual accuracy. We demonstrate the effectiveness of our proposed method across four tasks: an image captioning task with VLMs, two open-ended VQA tasks, and a VQA-as-judge task, each evaluated on distinct datasets and question types using LMMs.

## 2 RELATED WORK

### 2.1 BIAS IN IMAGE-TO-TEXT GENERATION

Image captioning and VQA involve generating textual descriptions for images. Prior studies (Fraser & Kiritchenko, 2024; Sathe et al., 2024; Howard et al., 2024b;a; Girrbach et al., 2025) have highlighted the presence of bias in such image-to-text tasks as detailed in Section 3. While these studies effectively quantify biases in model outputs, most remain limited to observational analysis and do not propose

concrete debiasing strategies. Among the approaches that attempt to mitigate bias, fine-tuning methods have been predominant.

## 2.2 DEBIASING VLMs AND LMMs

Fine-tuning-based debiasing has been explored for both image captioning (Hirota et al., 2023) and VQA (Park et al., 2020; Howard et al., 2024b; Yang et al., 2024; Girrbach et al., 2025). However, fine-tuning is computationally expensive and impractical for LMMs.

To avoid retraining, post-hoc methods have been proposed. Model-editing techniques (Wang et al., 2024) modify representations but rely on predefined anti-stereotypical knowledge. CLIP-clip (Wang et al., 2021), DeAR (Seth et al., 2023), model steering (Ratzlaff et al., 2024), and SFID (Jung et al., 2024) adjust frozen embeddings without altering the entire model. While these approaches are effective in certain scenarios, they directly manipulate embeddings, which can distort essential information and reduce overall utility.

While logit adjustment has been explored in methods like VDD (Zhang et al., 2024) to improve VQA performance, its mechanism and goals differ significantly from our work. VDD operates by subtracting a reference logit (derived from a meaningless or empty input) to cancel out the model's unconditional output biases, thereby reducing hallucinations. However, this technique was not designed for targeted social bias mitigation in generative tasks and, as our experiments show, has limited effectiveness for this purpose. In contrast, our approach introduces a dynamic adjustment that directly steers logits based on the real-time, measured misalignment between image attributes and the generated text, a mechanism specifically designed for debiasing.

## 3 PROBLEM DEFINITION

### 3.1 BIAS IN IMAGE CAPTIONING WITH VLMs

Image captioning generates descriptive text from an image using VLMs such as CLIP-CAP (Mokady et al., 2021) and BLIP (Li et al., 2022). A key fairness concern arises when an attribute identified in the generated caption does not align with that of the subject in the image (Hirota et al., 2023). For instance, given an image of a *female firefighter*, a profession stereotypically associated with men, the model might erroneously refer to the individual as "he," despite clear visual evidence to the contrary. This discrepancy suggests that VLMs can exhibit bias by associating certain professions or activities more frequently with specific attributes. While this type of image-text mismatch can apply to any attribute, we focus on gender bias as a representative case for this task.

**Evaluation Metric.** To quantify gender-related fairness issues, we evaluate the gender mismatch rate by detecting pronouns in the generated captions defined in (Jung et al., 2024). Given an image index $k$ in the test set, the mismatch indicator function is defined as follows

$$I_k = \begin{cases} 1 & \text{if (original gender)} \neq \text{(detected gender)} \\ 0 & \text{if (original gender)} = \text{(detected gender)} \quad \text{or} \quad \text{(neutral detected gender)} \end{cases}$$

where the misclassification rates for different gender groups are computed as $MR_{\mathcal{M}} = \frac{1}{|\mathcal{M}|} \sum_{k \in \mathcal{M}} I_k$, $MR_{\mathcal{F}} = \frac{1}{|\mathcal{F}|} \sum_{k \in \mathcal{F}} I_k$, and $MR_{\mathcal{O}} = \frac{1}{|\mathcal{O}|} \sum_{k \in \mathcal{O}} I_k$, with $\mathcal{M}$, $\mathcal{F}$, and $\mathcal{O}$ denote male, female, and overall, respectively. Instead of relying solely on the overall misclassification rate, we employ the Composite Misclassification Rate defined in (Jung et al., 2024), $MR_C = \sqrt{MR_{\mathcal{O}}^2 + (MR_{\mathcal{F}} - MR_{\mathcal{M}})^2}$, which captures both the overall error and the discrepancy between gender-specific error rates.

While debiasing the generated captions, we must also maintain their overall quality. To evaluate caption quality, we adopt MaxMETEOR and MaxSPICE following (Jung et al., 2024). The details are introduced in Appendix B.1. In evaluating image captioning models, a lower $MR_C$ indicates better fairness, while higher MaxMETEOR and MaxSPICE scores reflect improved captioning performance.

### 3.2 BIAS IN VISUAL QUESTION ANSWERING WITH LMMs

Bias is not limited to task-specific models; it can also be prevalent in more general LMMs. To quantify bias in LMMs, we consider two scenarios of VQA tasks with open-ended questions.

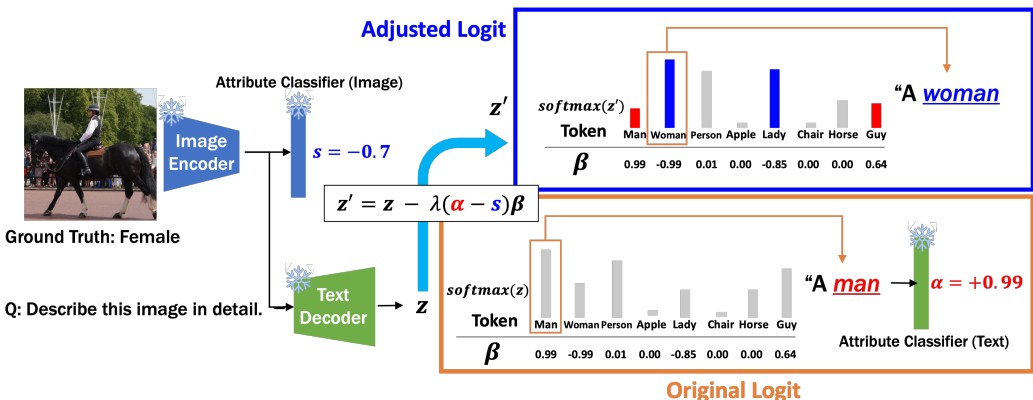

Figure 2: **Adaptive Logit Adjustment (ALA) for Bias Alignment** first generates next text token without modification. Then, it computes the target bias $s \in [-1, 1]$ from the frozen image representation and the bias score $\alpha(\mathbf{z}^t) \in [-1, 1]$ from the generated text by utilizing attribute classifier for image and text, respectively. If a discrepancy between $\alpha(\mathbf{z}^t)$ and $s$ is detected, the predicted logit vector is adjusted proportionally to the discrepancy. Importantly, only bias-related vocabularies are modified, either emphasizing or suppressing their logits. The direction and strength of the adjustment are precomputed as $\beta \in \mathbb{R}^V$, derived via gradient-based importance analysis (i.e., Integrated Gradients (Sundararajan et al., 2017)), ensuring targeted and interpretable debiasing.

**VQA-Task-1.** First, similar to image captioning, LMMs can generate biased responses when describing a given image with an open-ended question: `"Describe the photo in detail."` (Ratzlaff et al., 2024). The same fairness evaluation metric, $MR_C$, is used for this task, meaning that the generated text should contain pronouns that are either neutral or match the gender in the image.

**VQA-Task-2.** Second, we consider a more diverse scenario beyond gender bias. An LMM might be biased to generate harmful or toxic keywords for certain attributes when describing the image. For this task, we consider gender, physical traits and race, using the following prompt: `"What are the five keywords that describe the characteristics of people like the person in this image?"` as suggested in (Howard et al., 2024a). Ideally, the level of toxicity should be similar across attributes and their intersectional combinations. We use the average toxicity level, as measured by an external classifier, as our evaluation metric. The evaluation metric is defined as $D_{mean}$, while its details are introduced in Appendix B.2. In our evaluation, we use VQA-Task-1 and VQA-Task-2 to measure fairness, and introduce a third VQA task in Section 5.1 to assess the impact of our debiasing method on the model's core utility.

## 4 PROPOSED METHOD

In this section, we introduce *Adaptive Logit Adjustment* for Bias Alignment **(ALA-BA)** and Neutralization **(ALA-N)**, a post-hoc logit manipulation approach designed to debias image-to-text generation in both VLMs and LMMs.

Our approach operates by quantifying the attribute mismatch between the input image and the generated text during the autoregressive process. At each generation step $t$, the model's final layer outputs a logit vector $\mathbf{z}^t = (z_1, \ldots, z_V) \in \mathbb{R}^V$. To measure this bias, we leverage two pre-trained classifiers. First, an *image classifier*, $f^{\text{image}} : \mathbb{R}^d \to [-1, 1]$, processes the input image $x$ to produce a sensitive-attribute signal, $s = f^{\text{image}}(x)$, which serves as the *target bias*. Second, a *text classifier*, $f^{\text{text}} : \mathbb{R}^d \to [-1, 1]$, predicts the sensitive-attribute level in the generated text, from which we define the text's bias score as $\alpha(\mathbf{z}^t) = f^{\text{text}}(\mathbf{z}^t)$.

Ideally, we want $\alpha(\mathbf{z}^t) \approx s$, so that the model's textual bias aligns with the image-based bias. A large value of $|\alpha(\mathbf{z}^t) - s|$ therefore implies a significant misalignment between the image and the text.

## 4.1 ADAPTIVE LOGIT ADJUSTMENT (ALA)

Our goal is to push $\alpha(\mathbf{z}^t)$ closer to the target bias $s$. To achieve this, we consider a small update $\Delta\mathbf{z}^t$ and use a first-order Taylor expansion to approximate the change in $\alpha$,

$$\alpha(\mathbf{z}^t + \Delta\mathbf{z}^t) \approx \alpha(\mathbf{z}^t) + \sum_{i=1}^{V} \frac{\partial\alpha(\mathbf{z}^t)}{\partial z_i^t} \Delta z_i^t. \tag{1}$$

By subtracting $s$ from each side, we get

$$\left(\alpha(\mathbf{z}^t + \Delta\mathbf{z}^t) - s\right) \approx \left(\alpha(\mathbf{z}^t) - s\right) + \sum_{i=1}^{V} \frac{\partial\alpha(\mathbf{z}^t)}{\partial z_i^t} \Delta z_i^t. \tag{2}$$

Since our objective is to reduce the absolute discrepancy $|\alpha(\mathbf{z}^t) - s|$, a natural approach is to use a gradient-descent-like update on $\mathbf{z}^t$. We adjust each logit $z_i^t$ proportionally to the gradient $\frac{\partial\alpha(\mathbf{z}^t)}{\partial z_i^t}$, ensuring that $\alpha(\mathbf{z}^t)$ moves toward $s$ in each step. Thus, we design,

$$\Delta z_i^t = z_i^{t,\prime} - z_i^t = -\lambda\left(\alpha(\mathbf{z}^t) - s\right)\frac{\partial\alpha(\mathbf{z}^t)}{\partial z_i^t}, \tag{3}$$

where $z_i^{t,\prime}$ is the adjusted logit, and $\lambda > 0$ is a hyperparameter controlling the adjustment strength.

**Insight from Eq. 3:** Substituting Eq. 3 into Eq. 1, we obtain

$$\Delta\alpha = \alpha\left(\mathbf{z}^t + \Delta\mathbf{z}^t\right) - \alpha(\mathbf{z}^t) \approx \sum_{i=1}^{V} \frac{\partial\alpha(\mathbf{z}^t)}{\partial z_i^t} \Delta z_i^t$$

$$= \sum_{i=1}^{V} \frac{\partial\alpha(\mathbf{z}^t)}{\partial z_i^t}\left[-\lambda\left(\alpha(\mathbf{z}^t) - s\right)\frac{\partial\alpha(\mathbf{z}^t)}{\partial z_i^t}\right] = -\lambda\left(\alpha(\mathbf{z}^t) - s\right)\sum_{i=1}^{V}\left(\frac{\partial\alpha(\mathbf{z}^t)}{\partial z_i^t}\right)^2. \tag{4}$$

This formulation ensures that if $\alpha(\mathbf{z}^t) > s$, the update will decrease $\alpha(\mathbf{z}^t)$, and if $\alpha(\mathbf{z}^t) < s$, the update will increase $\alpha(\mathbf{z}^t)$, closing the gap. The magnitude of the update is controlled by the squared gradient norm $\sum_{i=1}^{V}(\frac{\partial\alpha(\mathbf{z}^t)}{\partial z_i^t})^2$, ensuring a stronger adjustment when $\alpha(\mathbf{z}^t)$ deviates significantly from $s$. This aligns $\alpha(\mathbf{z}^t)$ with $s$, ensuring that the model's textual attribute moves toward the image-based attribute or a neutralized target. The overall structure of the proposed ALA is illustrated in Figure 2.

## 4.2 BIASED TOKEN IDENTIFICATION

Because the partial derivatives $\frac{\partial\alpha(\mathbf{z}^t)}{\partial z_i^t}$ include the decoding process (i.e., selecting $\arg\max_i z_i^t$ to determine the next token), they are difficult to compute at each step. Instead, we approximate these gradients with token-specific importance scores $\beta_i \approx \frac{\partial\alpha(\mathbf{z}^t)}{\partial z_i^t}$, where $\beta = (\beta_1, \cdots, \beta_V) \in \mathbb{R}^V$. To identify tokens that significantly contribute to bias, we leverage gradient-based explanation techniques (Wang & Wang, 2022; Hao et al., 2021; Janizek et al., 2021). Specifically, for each token $i$ in the vocabulary, we compute a bias-related score $\beta_i$ measuring its contribution to the predicted sensitive attribute with the classifier $f^{\text{text}}$. Specifically, we take average over the gradient of the classifier's output with respect to the token embedding $e_i$ (Sundararajan et al., 2017). Although computing $\beta_i$ at every generation step is expensive, we can pre-compute a dictionary $\{\beta_i : i = 1, \ldots, V\}$ and store these values. The resulting fixed scores $\beta_i \in [-1, 1]$, normalized for consistency, serve as indicators of each token's inherent bias. Then, we rewrite Eq. 3 as

$$z_i^{t,\prime} = z_i^t - \lambda\left(\alpha(\mathbf{z}^t) - s\right)\beta_i, \tag{5}$$

and use $\beta_i$ in the logit adjustment step to steer the logit distribution toward the desired bias alignment.

However, applying logit adjustment at every time step may be computationally expensive due to the need for the text classifier $f^{\text{text}}$ to compute $\alpha(\mathbf{z}^t)$. Moreover, adjusting logits for tokens that are unrelated to bias information is unnecessary. To address this, we propose a selective logit adjustment strategy, where adjustment is applied only when the importance of the selected token $i_t$ at time $t$ is sufficiently high, i.e., $|\beta_{i_t}| \geq \tau$. We select $\tau = 0.1$ throughout the experiments based on analysis depicted in Figure 3. The detailed process of ALA is introduced in Algorithm 1.

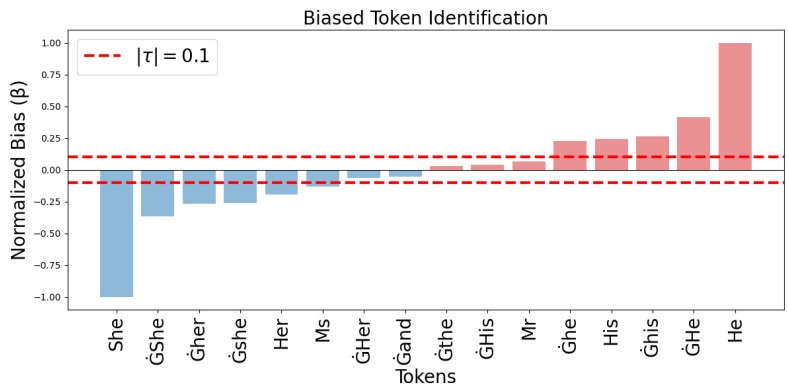

Figure 3: Selection of the threshold ($\tau$) for biased token identification. The normalized importance score ($\beta$) is analyzed for each token to assess its contribution to gender bias. The results indicate that setting $|\tau| = 0.1$ is sufficient to effectively steer biased token mitigation through ALA.

### 4.3 ALA FOR NEUTRALIZATION

In ALA-BA, $s \in [-1, 1]$ represents the target bias, guiding text generation by minimizing the discrepancy between $\alpha(\mathbf{z}^t)$ and $s$. However, users might prefer a neutralized output rather than one with aligned bias. ALA can be adapted for this purpose by minimizing the absolute bias score $|\alpha(\mathbf{z}^t)|$, ensuring that sensitive attributes are neither emphasized nor suppressed.

To achieve this, we modify the logit adjustment strategy by setting the target bias $s = 0$ and ensuring the update always reduces the magnitude of the bias score, $|\alpha(\mathbf{z}^t)|$. This adjustment mitigates tokens that contribute most to bias, regardless of whether they reflect positive or negative associations. As a result, the presence of sensitive attributes in the generated text is effectively reduced.

---

**Algorithm 1** Adaptive Logit Adjustment for Bias Alignment

---

**Require:** Input image $x$, VLM (or LMM) $F$ with its image encoder $G$, Input prompt $\mathcal{P}$, Pre-trained classifiers: $f^{image}$, $f^{text}$, Token bias score vector $\beta \in \mathbb{R}^V$, Maximum token length: max_token, Hyperparameter $\lambda$

**Ensure:** Debiased (or bias-aligned) text $\mathcal{T}$

1: $s \leftarrow f^{image}(G(x))$      // Target bias from image classifier
2: $\mathcal{T} \leftarrow []$      // Initialize output text as empty
3: **for** $t \leftarrow 1$ to max_token **do**
4:      $\mathbf{z}^t \leftarrow F(x, \mathcal{P}, \mathcal{T})$      // Obtain logits for next token based on partial text
5:      $i_t \leftarrow \arg\max_i \mathbf{z}_i^t$      // Choose the next token using the original logits
6:      **if** $|\beta_{i_t}| \geq \tau$ **then**
7:          $\alpha(\mathbf{z}^t) \leftarrow f^{text}(\mathcal{T} \cup \{i_t\})$      // Measure bias in current partial text
8:          $\mathbf{z}^{t,\prime} \leftarrow \mathbf{z}^t - \lambda(\alpha(\mathbf{z}^t) - s)\beta$      // Adaptive Logit Adjustment
9:          $i_* \leftarrow \arg\max_i \mathbf{z}^{t,\prime}$      // Choose the next token using the adjusted logits
10:      **else**
11:          $i_* \leftarrow i_t$ // If the next token is not significant for bias, skip the logit adjustment
12:      **end if**
13:      $\mathcal{T} \leftarrow \mathcal{T} \cup \{i_*\}$      // Append new token to the text sequence
14: **end for**

---

### 4.4 INTERSECTIONAL DEBIASING

Real-world debiasing often requires addressing multiple protected attributes simultaneously (e.g., race and gender). A key advantage of ALA is its signal agnosticism; it can ingest debiasing signals from diverse sources (e.g., neural classifiers and rule-based detectors) and aggregate them into a single logit adjustment via an Intersectional Logit Processor. We formulate this by summing the

adjustments from independent sources. For instance, to simultaneously mitigate gender and racial biases, the adjusted logit vector is computed as:

$$\mathbf{z}' = \mathbf{z} - \lambda_{\text{gender}}(\alpha_{\text{gender}} - s_{\text{gender}})\beta_{\text{gender}} - \lambda_{\text{race}}(\alpha_{\text{race}} - s_{\text{race}})\beta_{\text{race}}. \tag{6}$$

This formulation allows ALA to explicitly address complex, intersectional biases at inference time. Further implementation details, along with comprehensive quantitative evaluations and qualitative examples, are provided in Appendix O.

## 5 EXPERIMENTAL DETAILS

### 5.1 IMPLEMENTATION DETAILS

**Image Captioning.** We exclude images that contain multiple individuals to avoid ambiguity in gender identification. We evaluate two image captioning models, CLIP-CAP (Mokady et al., 2021) and BLIP (Li et al., 2022) using the MS-COCO dataset (Chen et al., 2015), which contains 10,780 images, each with five reference captions.

**VQA-Task-1.** We utilize the FACET (Gustafson et al., 2023) dataset, a real-world dataset containing gender/racial attributes, which makes it well suited for evaluating bias in LMMs. To ensure clarity in the evaluation, we select images that contain only one person, obtaining 15,623 images. The same fairness evaluation metric is adopted as in the image captioning task.

**VQA-Task-2.** We utilize the SocialCounterfactuals dataset (Howard et al., 2024b;a) to assess stereotypical bias in LMMs. This dataset comprises balanced synthetic images representing various intersectional attributes, including physical traits (skinny, obese, young, old, tattooed), gender (female, male), and race (Asian, Black, Indian, Latino, Middle Eastern, White). From more than 170k images, we select 5,200 by choosing 100 counterfactual sets for each intersectional bias combination (physical-gender, physical-race, and race-gender) to ensure a balance across attributes.

**VQA-Task-3 (Utility Preservation as a Judge).** To measure how debiasing affects a model's core utility, we design a "judge" task. This experiment tests whether a model can still accurately identify an attribute after debiasing—a scenario where methods that simply "blind" or remove bias-related information would likely fail. We use the FACET dataset and prompt the model with the following direct question: `"What is the gender of the person in this image? Choose either Male or Female as your response"`. The expectation is that the model should correctly identify the attribute without refusal. To quantify any harm to this capability, we define the "Worst-Case Accuracy Degradation" ($D_{WCA}$) as:

$$D_{WCA} = \min_{G \in \{\text{Female, Male}\}} (Acc(M_d, G) - Acc(M_o, G)),$$

where $M_d$ is the debiased model, $M_o$ is the original model, and a value closer to zero indicates better utility preservation.

**Summarization.** For clarification, we provide a table summarizing the model, dataset, and prompt used in each experiment in Table 2 in Appendix C.

### 5.2 APPLYING ALA FOR EACH TASK

The objective of each task differs. In image captioning and VQA-Task-1, both bias alignment (ALA-BA) and neutralization (ALA-N) are acceptable goals. For VQA-Task-2, however, the primary goal is to ensure non-toxicity across all attributes; we achieve this by setting the target bias to $s = -1$, treating non-toxicity as a specific form of bias alignment. Finally, the VQA-Task-3 judge task requires only bias alignment (ALA-BA), as the model must correctly identify the attribute in the image. For each VQA task, we utilize LLaVA-1.5 (Liu et al., 2024) and PaliGemma (Beyer et al., 2024), two prominent and powerful LMMs.

Table 1 summarizes the different experimental settings of ALA for each task. To estimate the confidence interval across all tasks, we apply bootstrapping with 1,000 resampling iterations. Furthermore, to verify that our approach scales to modern, instruction-tuned architectures, we extended our evaluation to Qwen2.5-VL-3B-Instruct Team (2025). Our results demonstrate that ALA effectively mitigates bias even in models with strong instruction-following capabilities, where standard prompt engineering fails. Detailed results and analysis are provided in Appendix L.

Table 1: ALA can be adapted to various scenarios by adjusting its configuration on target bias $s$, token bias $\beta$, and bias score in text $\alpha(\mathbf{z}^t) = f^{\text{text}}$.

| Configuration | ALA-BA | | | ALA-N |
|---|---|---|---|---|
| | Image Captioning | VQA Task 1 & 3 | Task 2 | |
| Target bias $s$ | $f^{\text{image}}$ | $f^{\text{image}}$ | -1 | 0 |
| Token bias | $\beta$ | $\beta$ | $\beta$ | $\|\beta\|$ |
| Bias score in text | $\alpha(\mathbf{z}^t)$ | $\alpha(\mathbf{z}^t)$ | $\alpha(\mathbf{z}^t)$ | $\|\alpha(\mathbf{z}^t)\|$ |

## 5.3 Pretraining External Classifiers

**Dataset:** We utilize the FairFace (Karkkainen & Joo, 2021) and Bias-in-Bios (De-Arteaga et al., 2019) datasets to pretrain $f^{\text{image}}$ and $f^{\text{text}}$, respectively, to mitigate gender bias in VLMs and LMMs. For toxicity debiasing, we use the Wikipedia Toxicity dataset (Thain et al., 2017). Using datasets for pretraining that are distinct from those used in our evaluations (COCO, FACET, and SocialCounterfactuals) demonstrates the transferability of our debiasing method.

**Architecture:** For $f^{\text{image}}$, we employ a logistic regression on frozen representations extracted by the target model's image encoder, e.g., CLIP (Radford et al., 2021). For $f^{\text{text}}$, we adopt a transformer-based classifier (Vaswani et al., 2017) to predict gender using the Bias-in-Bios dataset or toxicity using the Wikipedia Toxicity dataset. The $f^{\text{text}}$ classifier serves two purposes: (1) identifying biased tokens $\beta$, as described in Sec. 4.2, and (2) computing the bias score $\alpha(\mathbf{z}^t)$ in the generated text, as discussed in Sec. 4.1.

## 5.4 Comparison Methods

We compare ALA against several debiasing methods, including CLIP-clip (Wang et al., 2021), DeAR (Seth et al., 2023), and SFID (Jung et al., 2024), which primarily aim to mitigate bias in the representation space. We also implement VDD (Zhang et al., 2024), which applies logit adjustment to improve VQA performance. Moreover, we compare against a prompt engineering baseline, which uses an external classifier to add attribute-specific instructions to the prompt. Further details for each comparison are provided in Appendix D.

## 6 Result Analysis

### 6.1 Visual Analysis of Fairness-Utility Trade-offs

We visualize the trade-off between fairness and utility in Figure 4. Figure 4(a) demonstrates this trade-off for image captioning (fairness vs. caption quality), while Figure 4(b) and (c) show the results for VQA-Task-1 and VQA-Task-2, respectively. For these VQA plots, the y-axis uses the "Worst-Case Accuracy Degradation" metric from VQA-Task-3 to measure the impact on utility.

In these plots, the ideal method is located in the top-left quadrant, representing high fairness (a lower x-axis value) and minimal performance degradation (a higher y-axis value). As shown across all subfigures, our proposed methods, ALA-BA and ALA-N, are positioned near the top of the plot. This visually confirms that ALA preserves accuracy across gender subgroups. In contrast, other methods such as DeAR and CLIP-clip exhibit significant negative y-axis values, indicating that their fairness improvements come at the cost of performance degradation for at least one subgroup. Furthermore, ALA achieves these results while securing top-tier fairness scores, demonstrating a superior balance compared to competing approaches. Collectively, these visualizations underscore a primary contribution of our work: across different models, tasks, and fairness metrics, ALA consistently provides an effective solution that mitigates bias while preserving the model's essential utility (top-left). More detailed quantitative results are provided in Tables 5, 6, 7, and 8 in Appendix I.

**Applicability to Reasoning Tasks.** Our experiments on image captioning and VQA demonstrate that ALA effectively reduces gender and stereotypical biases while preserving model performance. Furthermore, we demonstrate that ALA's utility extends beyond generative tasks to discriminative

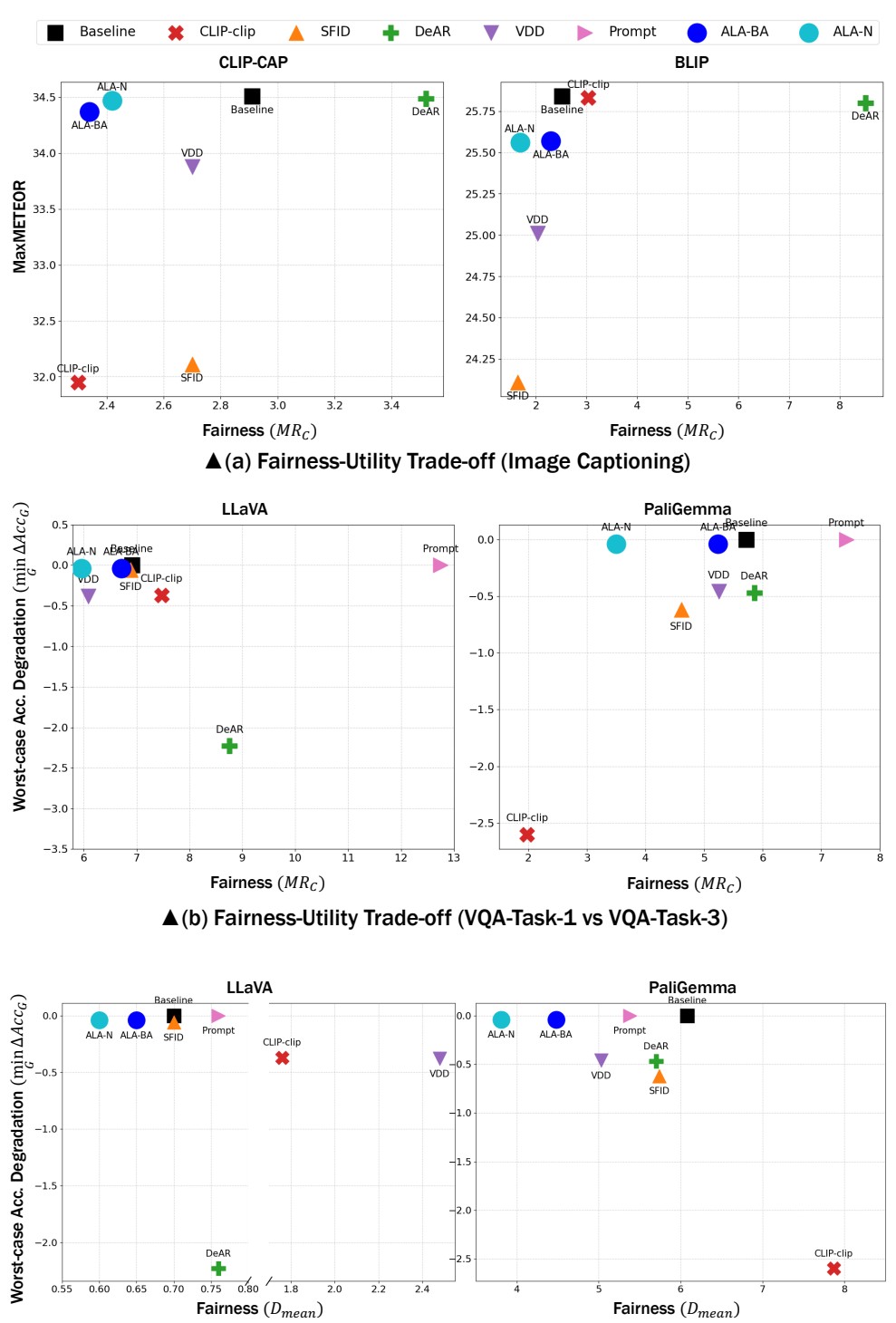

Figure 4: The fairness-utility trade-off of various debiasing methods across different models and tasks. **(a)** plots caption quality (MaxMETEOR) against gender fairness ($MR_C$) for image captioning models. **(b)** and **(c)** plot utility preservation ($D_{WCA}$) against fairness metrics ($MR_C$ for gender and $D_{mean}$ for stereotypes, respectively) for VQA tasks with LMMs.

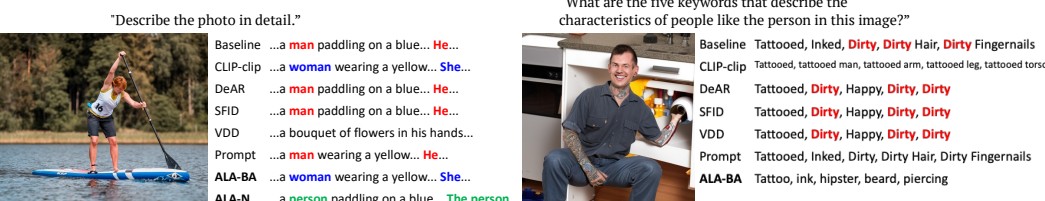

Figure 5: Qualitative examples demonstrating how ALA mitigates both attribute misalignment and stereotypical biases. Biased terms are highlighted in **red**, aligned corrections in **blue**, and neutralized outputs in **green**. In the left subfigure, **ALA-BA** adjusts output to align with the subject's attributes (e.g., changing "woman" to "man"), while **ALA-N** generates a neutral description (e.g., "a person"). The right subfigure illustrates the reduction of stereotypical bias, where ALA replaces negative keywords like "Dirty" with more objective descriptions.

reasoning tasks, such as occupation recognition, as detailed in Appendix M. It achieves the best or near-best fairness results across multiple tasks.

## 6.2 ABLATION STUDY AND LIMITATIONS

In ALA, the strength of logit adjustment is controlled by the hyperparameter $\lambda$. Our ablation study, detailed in Appendix F, shows that even a small adjustment (e.g., $\lambda = 0.1$) improves fairness, while $\lambda = 2$ provides the best trade-off between utility and fairness. However, excessively large values of $\lambda$ can degrade both performance and fairness, as shown in Figure 9 in the appendix.

Our method has two primary limitations. First, the effectiveness of ALA is dependent on the performance of the external attribute classifiers. We provide a theoretical analysis of this dependency in Appendix G. Second, the use of these classifiers introduces a minor computational overhead. However, this overhead is minimal: ALA incurs only a 3.1% increase in GPU utilization and a 1.2% increase in inference time. These costs are comparable to those of competing methods like CLIP-clip, SFID, and DeAR, remaining approximately twice as fast as VDD, which has a notably higher inference time. Furthermore, ALA's minimal overhead offers a vast efficiency advantage over model steering methods, which require a prohibitive full backward pass through the massive LMM decoder at every inference step. A more detailed analysis of these computational costs is provided in Appendix H.

## 7 CONCLUSION

We introduce Adaptive Logit Adjustment (ALA), a post-hoc debiasing method that refines token probabilities during autoregressive text generation. Unlike existing approaches that modify encoder or decoder representations, ALA directly adjusts logits to mitigate biases without distorting essential model outputs. ALA leverages external classifiers to detect bias misalignment between images and text, applies gradient-based importance analysis to identify biased tokens, and dynamically adjusts token probabilities to align the attributes of the input image and generated text. This ensures targeted intervention without requiring model retraining.

Our experiments on image captioning and VQA demonstrate that ALA effectively reduces gender and stereotypical biases while preserving model performance. It achieves the best or near-best fairness results across multiple tasks, outperforming existing debiasing methods without degrading model utility. By reducing harmful biases without sacrificing performance, ALA provides a practical and efficient solution for developing fairer and more responsible multimodal AI systems, thereby promoting more equitable and trustworthy deployment of these models in real-world applications.

ETHICS STATEMENT

This work is motivated by the goal of creating fairer and more responsible AI systems by mitigating biases in multimodal models. Our research exclusively uses publicly available, established benchmark datasets for experiments and for training the external classifiers, as detailed in Sections 5.1 and 5.3. No new data was collected, and no human subjects were involved. A key component of our methodology is the use of external attribute classifiers to guide the debiasing process. Recognizing that the performance of these classifiers is an important factor, we provide a detailed analysis in Appendix G which validates their accuracy and suitability for this framework. By proposing a method to reduce attribute misalignment and harmful stereotypes, we aim to contribute positively to the development of more equitable AI technology.

REPRODUCIBILITY STATEMENT

We are committed to ensuring the reproducibility of our research. The complete methodology for our proposed Adaptive Logit Adjustment (ALA) is detailed in Section 4, with a step-by-step implementation guide provided in Algorithm 1. All experimental setups, including the models, datasets, and prompts for each of the four tasks, are described in Section 5.1 and summarized in Table 2 (Appendix C). The training details for the external classifiers are provided in Section 5.3. The fairness and quality metrics used for evaluation are formally defined in Section 3.1 and Appendix B. To facilitate full verification of our results, our source code is included in the supplementary material.

ACKNOWLEDGEMENTS

This work was partially supported by the EMBRIO Institute, contract #2120200, a National Science Foundation (NSF) Biology Integration Institute, Purdue's Elmore ECE Emerging Frontiers Center, and NSF IIS #2146091, IIS #2345235.

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

## A    MORE QUALITATIVE RESULTS

Additional qualitative examples are provided in Figure 6, 7, and 8. Biased terms are highlighted in red, aligned corrections in blue, and neutralized outputs in green. The examples show that ALA-BA consistently mitigate the bias-misalignment issue between image and caption, while the ALA-N generates neutralized texts.

### CLIP-CAP, Attribute: Female

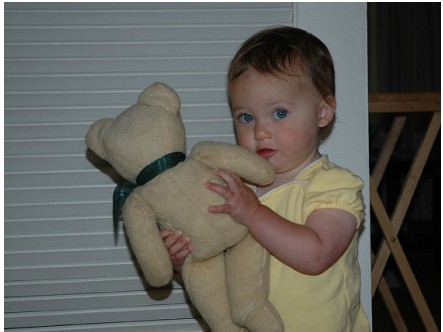

| | |
|---|---|
| Baseline | A baby holding a stuffed animal in **his** arms. |
| CLIP-clip | A baby holding a stuffed animal in **his** hands. |
| DeAR | A baby holding a stuffed animal in **his** arms. |
| SFID | A baby holding a teddy bear in **his** arms. |
| VDD | A baby holding a stuffed animal in **his** arms. |
| **ALA-BA** | A baby holding a stuffed animal in **her** arms. |
| **ALA-N** | A baby holding a stuffed animal in **it's** arms. |

### CLIP-CAP, Attribute: Female

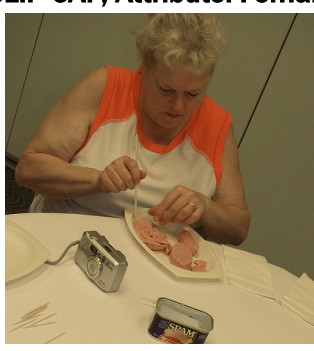

| | |
|---|---|
| Baseline | A **man** cutting up a piece of meat on a table. |
| CLIP-clip | A **woman** cutting up a piece of meat... |
| DeAR | A **man** cutting up a piece of meat on a table. |
| SFID | A **woman** in a white shirt cutting up a cake. |
| VDD | A **woman** cutting up carrots on a white plate. |
| **ALA-BA** | A **woman** cutting up a piece of meat on a table. |
| **ALA-N** | A **woman** cutting up a piece of meat on a table. |

### BLIP, Attribute: Male

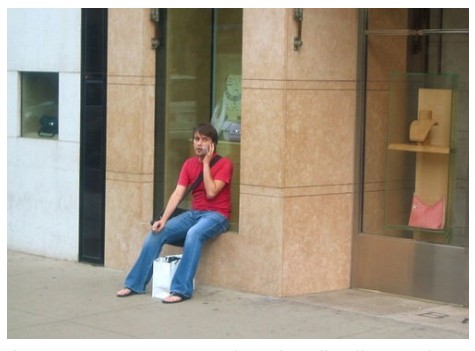

| | |
|---|---|
| Baseline | a **woman** sitting on the sidewalk talking on her cell |
| CLIP-clip | a **person** sitting on a sidewalk |
| DeAR | a **man** sitting on the sidewalk talking on a cell |
| SFID | a **person** on a cell phone |
| VDD | a **woman** sitting on the sidewalk talking on her cell |
| **ALA-BA** | a **man** sitting on the sidewalk talking on his cell |
| **ALA-N** | a **person** sitting on the sidewalk talking on a cell |

### BLIP, Attribute: Male

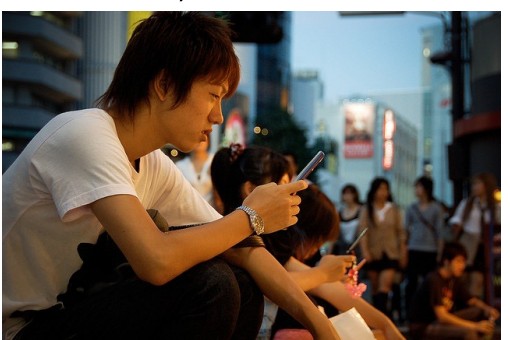

| | |
|---|---|
| Baseline | a **woman** sitting on a bench looking at her phone |
| CLIP-clip | a **woman** sitting on a bench looking at her phone |
| DeAR | a **man** sitting on a bench looking at his phone |
| SFID | a **person** on a cell phone |
| VDD | a **woman** sitting on a bench looking at her phone |
| **ALA-BA** | a **man** sitting on a bench with a cell in his hand |
| **ALA-N** | a **person** sitting on a bench looking at a cell |

Figure 6: Additional qualitative examples demonstrating ALA's effectiveness on image captioning models, CLIP-CAP and BLIP.

**LLaVA-1.5-7B, Attribute: Female**

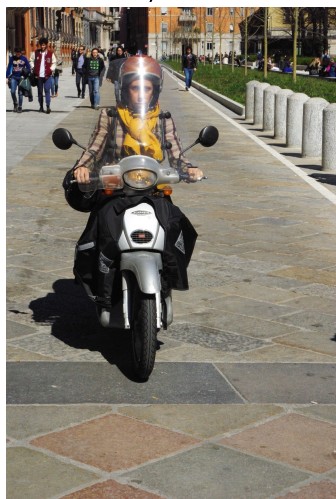

**LLaVA-1.5-7B, Attribute: Female**

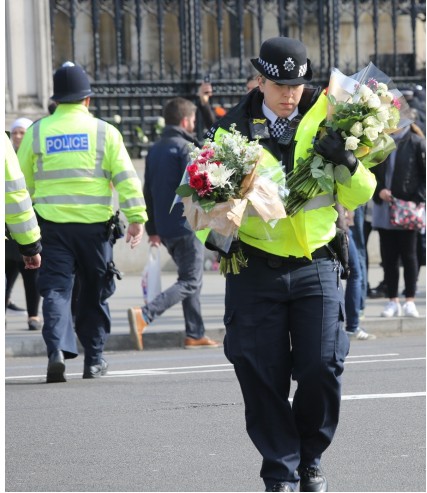

| | |
|---|---|
| Baseline | The image features a **man** riding a motorcycle down a sidewalk. **He** is wearing a yellow jacket and a helmet, ensuring **his** safety while riding. The motorcycle is positioned in the center of the scene, with... |
| CLIP-clip | The image depicts a **man** riding a scooter down a sidewalk. **He** is wearing a yellow jacket and a helmet, ensuring **his** safety while riding. The scooter is parked on the sidewalk... |
| DeAR | The image features a **man** riding a motorcycle down a sidewalk. **He** is wearing a yellow jacket and a helmet, ensuring **his** safety while riding. The motorcycle is positioned in the center of the scene, with... |
| SFID | The image features a **man** riding a motorcycle down a sidewalk. **He** is wearing a yellow jacket and a helmet, ensuring **his** safety while riding. The motorcycle is positioned in the center of the scene, with... |
| VDD | The image features a **person** riding a motorcycle down a street. The **person** is wearing a helmet and is positioned in the middle of the scene. |
| Prompt | The image features a **woman** riding a motorcycle down a sidewalk. **She** is wearing a yellow scarf and a black jacket, which adds a touch of style to **her** outfit. The motorcycle is positioned in the center of... |
| **ALA-BA** | The image features a **woman** riding a motorcycle down a sidewalk. **She** is wearing a yellow scarf and a black jacket, which adds a touch of style to **her** outfit. The motorcycle is positioned in the center of... |
| **ALA-N** | The image features a **person** riding a motorcycle down a sidewalk. The **person** is wearing a helmet and is positioned in the center of the scene. The motorcycle is parked on the sidewalk, and the rider appears... |

| | |
|---|---|
| Baseline | The image features a police officer walking down a street, holding a **bouquet of flowers in his hands.** The officer appears to be in a crowd, possibly during a parade or event. |
| CLIP-clip | The image depicts a police officer walking down a busy city street, **holding a bouquet of flowers.** The officer is wearing a yellow jacket and a hat, making them easily identifiable. |
| DeAR | The image features a police officer walking down a street, holding a **bouquet of flowers in his hands.** The officer appears to be in a crowd, possibly during a parade or event. |
| SFID | The image features a police officer walking down a street, holding a **bouquet of flowers in his hands.** The officer appears to be in a crowd, possibly during a parade or event. |
| VDD | The image depicts a police officer walking down a street, **holding a bouquet of flowers.** The officer is wearing a yellow jacket, which stands out against the surroundings. |
| Prompt | The image features a **man** wearing a yellow jacket and a police hat, walking down a street while holding a bouquet of flowers. **He** appears to be a police officer, possibly participating in a parade or event. |
| **ALA-BA** | The image features a police officer walking down a street, holding a **bouquet of flowers in her hands.** The officer appears to be in a crowd, possibly during a parade or event. |
| **ALA-N** | The image features a police officer walking down a street, holding a **bouquet of flowers in their hands.** The officer appears to be in a crowd, possibly during a parade or event. |

Figure 7: Additional qualitative examples demonstrating ALA's effectiveness on LLaVA-1.5-7B model.

# B  EVALUATION METRIC

## B.1  EVALUATION METRIC FOR IMAGE CAPTIONING

For evaluating the caption quality, we adopt METEOR (Banerjee & Lavie, 2005) and SPICE (Anderson et al., 2016). METEOR (Banerjee & Lavie, 2005) evaluates the trade-off between precision and recall of n-grams in generated captions while accounting for synonym matches. Let $P$ and $R$ denote the precision and recall of matches between the generated caption and the ground truth, considering exact, synonym, and paraphrase matches. METEOR is computed as:

$$\text{METEOR} = F_{\text{mean}} \cdot (1 - \text{Pen})$$

where

$$F_{\text{mean}} = \frac{10 \cdot P \cdot R}{R + 9 \cdot P}$$

represents a harmonic mean, and the penalty term is defined as:

**PaliGemma, Attribute: Female**

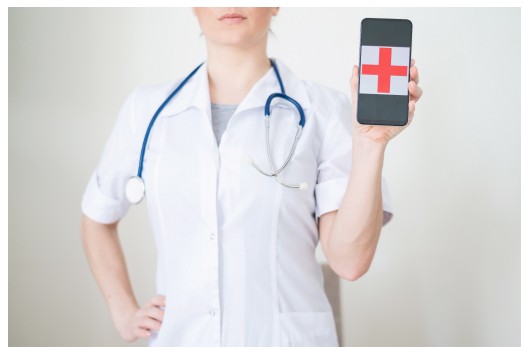

Baseline — In this image we can see **a person** wearing a stethoscope and holding a mobile phone **in his hand.**

CLIP-clip — **A woman** in a white panikle, with a blue eksternal mikrofon, is holding a cell phone with a red plus sign on it. The phone has a black case and a black and white button on the front.

DeAR — In this image we can see **a person** wearing a stethoscope and holding a mobile phone **in his hand**

SFID — In this image I can see **a person** wearing white color dress and stethoscope is holding a mobile phone **in the hand.**

VDD — In this image we can see a **woman** standing and holding a mobile phone **in her hand.** In the background there is wall.

Prompt — In this image we can see a **woman** wearing a white color coat and a stethoscope is holding a mobile phone **in her hand.**

**ALA-BA** — In this image we can see **a person** wearing a stethoscope and holding a mobile phone **in her hand.**

**ALA-N** — In this image we can see **a person** wearing a stethoscope and holding a mobile phone **in the hand.**

**PaliGemma, Attribute: Female**

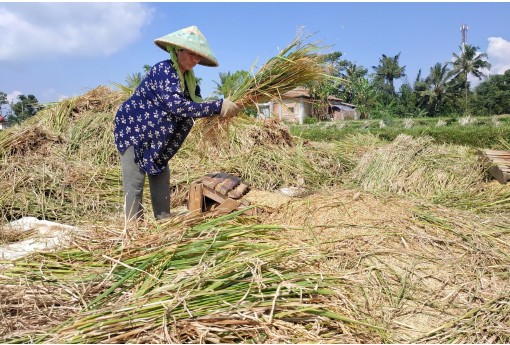

Baseline — In this image we can see **a person** standing on the ground and holding a bunch of grass **in his hand.** In the background we can see a house, trees, tower and the sky.

CLIP-clip — In this image, I can see **a person** is standing on the ground. I can see a few plants, a few trees, a house, a pole, a few people, a few clouds...

DeAR — In this image we can see **a person** wearing a hat and holding a bunch of grass **in his hand.** In the background, we can see a house, trees, a tower and the sky.

SFID — In this image I can see **a person** wearing blue color dress and hat is standing on the ground. I can see few dry grass, few trees, few buildings, few poles and the sky is in blue and white color.

VDD — In this image we can see **a person** standing and holding some objects **in his hands.** In the background we can see houses, trees, tower and sky with clouds.

Prompt — In this image we can see **a person** wearing a hat and holding a bunch of grass **in her hand.** In the background, we can see a house, trees, a tower and the sky.

**ALA-BA** — In this image we can see **a person** standing on the ground and holding a bunch of grass **in her hand.** In the background we can see a house, trees, tower and the sky.

**ALA-N** — In this image we can see **a person** standing on the ground and holding a bunch of grass **in the hands.** In the background, we can see a house, trees, a tower and the sky.

Figure 8: Additional qualitative examples demonstrating ALA's effectiveness on PaliGemma model.

$$\text{Pen} = 0.5 \times \left( \frac{\text{number of chunks}}{\text{number of matches}} \right)^3$$

A chunk refers to a sequence of consecutive words in the generated caption that appear in the reference.

SPICE (Anderson et al., 2016), on the other hand, assesses the semantic quality of captions by comparing sets of propositional semantic tuples extracted from both the candidate and reference captions. It is computed as the F1 score of precision and recall between these tuples, providing a measure of semantic alignment.

Following (Jung et al., 2024), the quality evaluation considers both the original ground-truth caption and a neutral alternative,

$$\text{MaxMETEOR} = \max(\text{METEOR}(T_{\text{truth}}, T_{\text{caption}}), \text{METEOR}(T_{\text{neutral}}, T_{\text{caption}})),$$
$$\text{MaxSPICE} = \max(\text{SPICE}(T_{\text{truth}}, T_{\text{caption}}), \text{SPICE}(T_{\text{neutral}}, T_{\text{caption}})).$$

## B.2 EVALUATION METRIC FOR VQA-TASK-2

For evaluation, we use a toxicity classifier $f^{\text{text}}$, trained on the Wikipedia Toxicity dataset (Thain et al., 2017), to score each of the five keywords generated for a given image $i$. The image's overall toxicity is the average of these five keyword scores.

We then compute the mean toxicity for each specific attribute $a$ within a broader category $G$ (e.g., physical traits, gender, or race) as follows:

$$\text{toxic}_a^G = \frac{1}{|I_a|} \sum_{i \in I_a} \text{mean}_{k \in \{1,\dots,5\}} \text{toxic}_{i,k},$$

where $I_a$ is the set of images associated with attribute $a$. To measure the fairness disparity within a category, we calculate $D_{\max}^G$, which is the maximum absolute difference in mean toxicity between any two attributes in that category:

$$D_{\max}^G = \max_{a,b \in G} \left| \text{toxic}_a^G - \text{toxic}_b^G \right|.$$

Finally, to create a single, comprehensive fairness score across all categories, we define $D_{\text{mean}}$ as the average of these maximum disparities:

$$D_{\text{mean}} = \frac{1}{|\mathcal{G}|} \sum_{G \in \mathcal{G}} D_{\max}^G,$$

where $\mathcal{G} = \{\text{physical traits, gender, race}\}$ is the set of all attribute categories considered. A lower $D_{\text{mean}}$ indicates greater fairness, as it signifies smaller toxicity gaps across all attributes.

## C  SUMMARY OF EXPERIMENTAL SETTING

The details of experimental settings are summarized in Table 2.

Table 2: Experimental Setting Summary

| Task | Image Captioning | VQA-Task-1 | VQA-Task-2 | VQA-Task-3 |
|---|---|---|---|---|
| Dataset | COCO-Caption | FACET | SocialCounterfactuals | FACET |
| Model | CLIP-CAP, BLIP | LLaVA-1.5-7B PaliGemma | LLaVA-1.5-7B PaliGemma | LLaVA-1.5-7B PaliGemma |
| Target Bias | Gender | Gender | Gender, Physical, Race | - |
| Prompt | - | Describe the photo in detail | What are the five keywords that describe the characteristics of people like the person in this image? | What is the gender of the person in this image? |
| Fairness Evaluation Metric | Composite Misclassification Rate ($MR_C$) | | Maximum Disparity ($D_{\text{mean}}$) | Accuracy & Worst-Case Accuracy Degradation ($D_{WCA}$) |

## D  DETAILS FOR COMPARISON METHOD

DeAR employs adversarial training by optimizing an adaptor network on the encoder's representations to deceive a sensitive attribute classifier, thereby eliminating bias-related information. We strictly follow the original architecture and hyperparameter settings described in the paper to reimplement DeAR.

CLIP-clip and SFID, on the other hand, focus on pruning biased features in the representation space. SFID can be applied to the encoder, decoder, or both by identifying bias-related features at each component and masking them. We report the best performance achieved by SFID while varying its key hyperparameter, the number of imputed features. As a special case, we adopt SFID as a bias-alignment baseline for comparison for VQA-Task-3, denoted SFID-BA. Further details are provided in the Appendix E. Although CLIP-clip was initially proposed to remove bias from the encoder's embeddings, (Jung et al., 2024) suggests that CLIP-clip can be extended to the decoder as well like SFID. CLIP-clip mitigates bias by removing specific features from the representation space, effectively reducing its dimensionality. However, this direct feature removal is incompatible with encoder-decoder architectures, as it alters the expected representation size. To address this

issue, we adapt CLIP-clip for image-to-text tasks using a zero-pruning strategy, which preserves the dimensionality while removing the biased components.

In contrast, VDD (Zhang et al., 2024) was originally designed to mitigate hallucination by adjusting the output logits through subtraction of a reference logit derived from an empty or meaningless image. We implement VDD and include it for all evaluation scenarios.

The prompt engineering strategy relies on the external classifier, which is used in ALA. In VQA-Task-1, in front of the input prompt, we put an instruction `"The gender of the person in the image is {PREDICTED GENDER}"`. to guide the generated text to align with the desired gender.

In the SocialCounterfactuals dataset for VQA-Task-2, intersectional bias arises from a combination of three categories: physical appearance, race, and gender. While comparable debiasing methods can address specific types of bias, CLIP-clip and SFID are primarily effective in mitigating bias within a single category. However, when multiple attributes interact to create intersectional bias in the test set, only DeAR is capable of addressing it. The prompt engineering method simply use an instruction, `"Do not include toxicity in the generated text."` To evaluate their debiasing performance, we report results where CLIP-clip and SFID are applied separately to mitigate bias in race and gender, the only attributes included in the FairFace debiasing training set, as shown in Table 7. In contrast, our method explicitly addresses this issue across different bias types by setting the target bias in stereotypical bias as $s = -1$, non-toxicity, as described in Table 1.

On the other hand, model steering (Ratzlaff et al., 2024) is not included in comparison as it requires computing the gradient of the LMM *w.r.t* the input image, which exceeds our available computational resources.

## E    BIAS ALIGNMENT WITH SFID (JUNG ET AL., 2024)

Selective Feature Imputation for Debiasing (SFID) (Jung et al., 2024) is designed to obscure bias-related information in the representation space. Specifically, it determines feature importance using a Random Forest classifier (Breiman, 2001) trained to predict sensitive attributes. It then imputes values in the most important features with those of the mean of low-confidence samples from the validation dataset, ensuring that all features resemble ambiguous (low-confidence) samples.

However, this method can be applied in a different direction. Instead of obscuring important features, they can be reinforced for certain demographic groups when a clear attribute signal is present, by leveraging high-confidence samples. We adopt this strategy for the VQA-Task-3 task and report the results of SFID-BA (Bias Alignment) in Table 8.

## F    ABLATION STUDY

In ALA, the strength of logit adjustment is controlled by the hyperparameter $\lambda$. To analyze its impact, we conduct ablation studies by varying $\lambda$ and evaluating its effect on both performance and fairness in image-to-text tasks.

For VLMs, we assess the effect of $\lambda$ using CLIP-CAP for both **Bias Alignment** and **Neutralization**, as shown in Figure 9 (a). The results indicate that while excessively large $\lambda$ can degrade both performance and fairness, an appropriately chosen $\lambda$, such as $\lambda = 2$, improves fairness without sacrificing performance. Notably, even a small adjustment, such as $\lambda = 0.1$, already leads to noticeable fairness improvements compared to the baseline. This demonstrates that ALA can effectively mitigate bias with minimal intervention, making it adaptable to scenarios with strict performance constraints.

For LMMs, we conduct a similar ablation study using the VQA task on the FACET dataset with LLaVA. Figure 9 (b) illustrates how the fairness metric $MR_C$ for the open-ended description task, VQA-Task-1, varies with different values of $\lambda$ for each model. Utility is measured separately using a different task, VQA-Task-3. Similar to the image captioning results in VLMs, fairness improves with moderate values of $\lambda$, such as 2, while excessively large values degrade both fairness and utility. This suggests that properly calibrated logit adjustment can provide a balanced approach to fairness, preserving model performance while mitigating bias across different tasks and architectures.

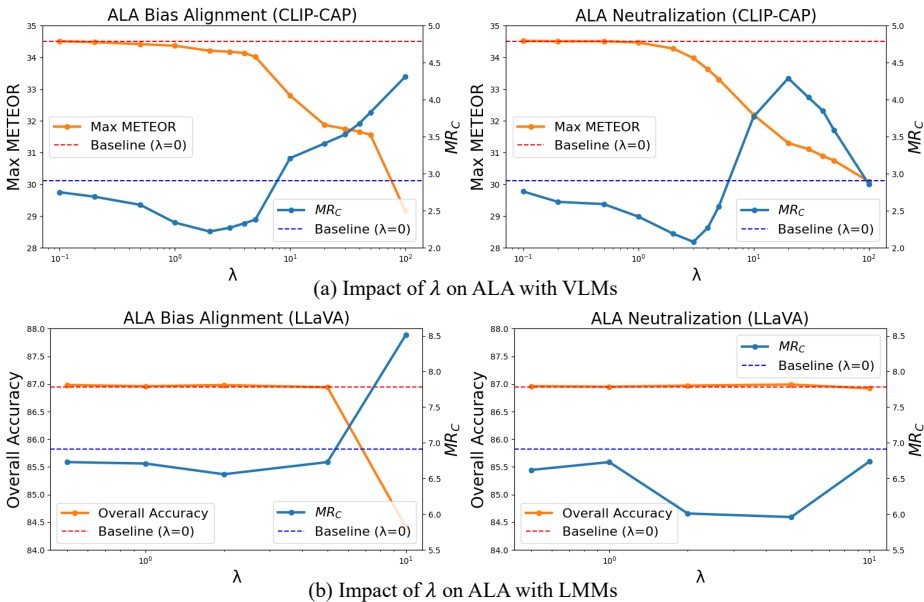

Figure 9: Impact of logit adjustment strength ($\lambda$) on VLMs for image captioning (CLIP-CAP) and LMMs for VQA tasks (LLaVA). The orange curves represent model performance (higher is better): MaxMETEOR score for image captioning and overall accuracy for VQA-as-judge. The blue curves denote fairness, $MR_C$ (lower is better). Moderate values of $\lambda$, such as $\lambda = 2$, improve fairness without degrading performance. Both Bias Alignment (left) and Neutralization (right) exhibit a similar trend, though Neutralization achieves slightly better fairness.

## G  CLASSIFIER PERFORMANCE ANALYSIS

The use of an external classifier to guide debiasing is a common strategy in post-hoc methods (Wang et al., 2022; Jung et al., 2024). A valid consideration for any such framework, including ours, is its reliance on the performance of this external classifier. Therefore, we conduct an analysis to ensure the classifiers used in our framework are sufficiently accurate and robust.

We train simple logistic regression classifiers on top of frozen image embeddings from our target models (LLaVA-1.5 and PaliGemma) using the FairFace dataset. This dataset is ideal for isolating demographic attributes, as it contains only facial images and minimizes confounding variables like background scenery or clothing.

As shown in Table 3, our classifiers achieve high accuracy (over 93%). More importantly, we analyzed the nature of their errors. The results show that a high proportion (over 78%) of misclassified samples were those the classifier deemed ambiguous (i.e., its prediction score $|s|$ was close to zero). This is a crucial finding; it indicates that the classifier's failures occur on genuinely difficult cases, not by making confident mistakes on clear ones. This behavior makes it highly suitable for our framework, as low-confidence scores from the classifier naturally result in smaller, more conservative logit adjustments, preventing overcorrection based on an uncertain signal and resulting in neutralization effect.

Finally, our framework is designed with an inherent safeguard against classifier error. The ALA-N (Neutralization) variant mitigates this reliance by setting the target bias to a neutral state ($s = 0$). This approach focuses on dampening any strong bias signal rather than aligning to a specific (and potentially misclassified) attribute, making it exceptionally robust to imperfect classifier predictions.

Table 3: Performance of gender attribute classifiers trained on frozen image embeddings from LLaVA and PaliGemma using the FairFace dataset. High accuracy and a large proportion of ambiguity in errors support their robustness for our framework.

| Frozen Encoder | Accuracy | Portion Ambiguous in Misclassified |
|---|---|---|
| LLaVA-1.5-7B | 96.37% | 78.39% |
| PaliGemma | 93.23% | 83.02% |

## H    COMPUTATIONAL COST ANALYSIS

As we adopt external image and text classifiers, we carefully examine the additional computational cost. Table 4 shows only a slight increase in RAM and GPU usage, as the external classifiers remain lightweight—a single-layer classifier for image inputs and a two-block transformer for text inputs. Notably, the increases are comparable across all comparison methods. However, VDD exhibits a substantially slower inference time, with a 101.5% increase, as it requires performing inference twice for each input, while our method incurs only a 1.2% increase.

Table 4: Resource consumption comparison of different methods.

| Method | CPU Memory (MB) | | RAM Usage (MB) | | GPU Memory (MB) | | Inference Time (s) | |
|---|---|---|---|---|---|---|---|---|
| | Value | % | Value | % | Value | % | Value | % |
| Baseline | 1368.48 | - | 69578.89 | - | 13481.79 | - | 1.5621 | - |
| CLIP-clip | 1630.69 | 19.2 | 69821.79 | 0.3 | 13873.67 | 2.9 | 1.5639 | 0.1 |
| SFID | 1634.55 | 19.4 | 69755.95 | 0.3 | 13873.67 | 2.9 | 1.5739 | 0.8 |
| DeAR | 1406.82 | 2.8 | 69593.04 | 0.0 | 13882.86 | 3.0 | 1.5767 | 0.9 |
| VDD | 1426.94 | 4.3 | 70022.26 | 0.6 | 13876.67 | 2.9 | 3.1472 | 101.5 |
| **Ours (ALA)** | 1615.74 | 18.1 | 70137.92 | 0.8 | 13894.22 | 3.1 | 1.5815 | 1.2 |

## I    DETAILED EXPERIMENTAL RESULTS

Tables 5, 6, 7, and 8 demonstrate the effectiveness of the proposed method, ALA-BA (Bias Alignment) and ALA-N (Neutralization). Specifically, ALA achieves the best or second-best fairness while minimizing accuracy loss, highlighting the minimal trade-off between utility and fairness. In image captioning (Table 5), ALA demonstrates strong fairness while maintaining caption quality. In the VQA open-ended question tasks (Tables 6, 7), ALA consistently achieves top fairness results while preserving accuracy in the VQA-as-judge task (Table 8), whereas representation-based debiasing approaches often degrade utility.

While these tables provide a granular breakdown of performance on each task, the fairness-utility trade-off can be more intuitively understood through visual analysis. For a comprehensive summary and a direct comparison of this trade-off across all methods, we refer the reader to Figure 4 and discussion in Section 6.1.

Table 5: Experimental results for image captioning on COCO-caption dataset. **Bold** indicates the best result for each baseline, while underline denotes the second and third-best result.

| **Image Captioning** | | Caption Quality | | Misclassification Rate | | |
|---|---|---|---|---|---|---|
| | | Max METEOR($\uparrow$) | Max SPICE ($\uparrow$) | \|Male-Female\|($\downarrow$) $(\|MR_\mathcal{M} - MR_\mathcal{F}\|)$ | Overall ($\downarrow$) $(MR_\mathcal{O})$ | **Composite** ($\downarrow$) $(\mathbf{MR}_\mathcal{C})$ |
| CLIP-CAP | Baseline | **34.51±0.20** | **25.38±0.18** | 2.08±0.72 | 2.00±0.28 | 2.91±0.59 |
| | CLIP-clip | 31.95±0.20 | 23.93±0.16 | **0.37±0.36** | 2.26±0.31 | **2.30±0.32** |
| | SFID | 32.11±0.17 | 24.03±0.18 | 1.41±0.64 | 2.25±0.26 | 2.70±0.44 |
| | DeAR | 34.49±0.21 | 25.35±0.17 | 2.87±0.74 | 2.06±0.29 | 3.52±0.66 |
| | VDD | 33.88±0.22 | 24.77±0.17 | 1.65±0.75 | 2.14±0.24 | 2.70±0.54 |
| | **ALA-BA** | 34.37±0.19 | 25.27±0.17 | 1.19±0.64 | **1.97±0.27** | 2.34±0.43 |
| | **ALA-N** | 34.47±0.21 | 25.35±0.18 | 1.34±0.70 | 1.99±0.28 | 2.42±0.44 |
| BLIP | Baseline | **25.84±0.13** | **18.58±0.13** | 2.11±0.62 | 1.38±0.21 | 2.52±0.60 |
| | CLIP-cli | 25.83±0.13 | 18.50±0.11 | 2.73±0.63 | 1.31±0.20 | 3.04±0.63 |
| | SFID | 24.11±0.16 | 18.13±0.13 | 1.45±0.47 | **0.77±0.16** | **1.65±0.47** |
| | DeAR | 25.80±0.14 | 18.41±0.12 | 8.09±0.97 | 2.62±0.31 | 8.51±1.00 |
| | VDD | 25.01±0.13 | 18.03±0.13 | 1.70±0.50 | 1.15±0.19 | 2.04±0.48 |
| | **ALA-BA** | 25.57±0.13 | 18.40±0.13 | 1.86±0.53 | 1.37±0.22 | 2.30±0.51 |
| | **ALA-N** | 25.56±0.13 | 18.42±0.13 | **1.39±0.47** | 0.91±0.18 | 1.69±0.43 |

Table 6: Experimental results for VQA open-ended question for bias misalignment on FACET dataset. **Bold** indicates the best result for each baseline, while underline denotes the second-best result.

| **VQA-Bias-1** | LLaVA-1.5 | | | PaliGemma | | |
|---|---|---|---|---|---|---|
| | $\|MR_\mathcal{M} - MR_\mathcal{F}\|$ | $MR_\mathcal{O}$ | $\mathbf{MR}_\mathcal{C}$ | $\|MR_\mathcal{M} - MR_\mathcal{F}\|$ | $MR_\mathcal{O}$ | $\mathbf{MR}_\mathcal{C}$ |
| Baseline | 3.07±1.18 | 6.14±0.48 | 6.91±0.75 | 3.51±1.07 | 4.44±0.41 | 5.72±0.84 |
| CLIP-clip | 3.82±1.29 | 6.33±0.47 | 7.48±0.84 | 2.12±0.81 | **2.93±0.66** | **1.98±0.27** |
| SFID | 2.97±1.18 | 6.10±0.44 | 6.89±0.70 | **1.03±0.92** | 4.45±0.39 | 4.61±0.45 |
| DeAR | 6.17±1.29 | 6.19±0.46 | 8.76±1.04 | 3.53±1.13 | 4.60±0.38 | 5.86±0.85 |
| VDD | 2.02±1.11 | **5.73±0.47** | 6.09±0.61 | 2.29±1.02 | 4.69±0.42 | 5.25±0.63 |
| Prompt | 6.47±1.54 | 10.94±0.63 | 12.75±1.54 | 5.18±1.16 | 5.25±0.44 | 7.43±0.96 |
| **ALA-BA** | 2.86±2.74 | 6.03±1.33 | 6.71±1.86 | 2.55±1.03 | 4.50±0.42 | 5.24±0.73 |
| **ALA-N** | **1.25±0.93** | 5.78±0.45 | **5.96±0.50** | 1.06±0.72 | 3.31±0.34 | 3.50±0.42 |

## J  COMPUTATIONAL RESOURCE

## K  QUANTITATIVE ANALYSIS OF LOGIT SPACE

To quantitatively analyze how each debiasing method alters the model's output distribution, we measured the entropy of the logits during text generation in VQA-Task-2, debiasing the toxicity. This analysis allows us to understand the mechanism behind each method's performance by examining the shape of the probability distribution over different sets of tokens.

### K.1  METHODOLOGY

For each token generation step, we first compute the probability distribution $P$ over the entire vocabulary by applying a softmax function to the logit vector $z$. The overall entropy is the Shannon entropy of this distribution, calculated as $H(P) = -\sum_i p_i \log p_i$.

For a more granular analysis, we partitioned the vocabulary into two disjoint sets based on pre-computed token bias scores ($\beta$): undesirable/toxic (High Bias, $\beta > 0.01$) and desirable/non-toxic (Low Bias, $\beta < -0.01$). We then computed the conditional entropy for each partition, which is the entropy calculated only over the renormalized probabilities of the tokens within that set.

### K.2  ANALYSIS OF RESULTS

Our quantitative analysis reveals the unique characteristics of our proposed method, ALA-BA. Table 10 shows that ALA-BA is a decisive intervention, achieving a low mean entropy (1.6713) that

Table 7: Experimental results for VQA open-ended question for stereotypical bias on SocialCounter-factuals dataset. **Bold** indicates the best result for each baseline, while underline denotes the second best result.

| VQA-Bias-2 | LLaVA-1.5 | | | | PaliGemma | | | |
|---|---|---|---|---|---|---|---|---|
| | $D^P_{\max}$ ($\downarrow$) | $D^R_{\max}$ ($\downarrow$) | $D^G_{\max}$ ($\downarrow$) | $D_{\text{mean}}$ ($\downarrow$) | $D^P_{\max}$ ($\downarrow$) | $D^R_{\max}$ ($\downarrow$) | $D^G_{\max}$ ($\downarrow$) | $D_{\text{mean}}$ ($\downarrow$) |
| Baseline | 1.07±0.18 | 0.64±0.17 | 0.40±0.13 | 0.70 | 8.62±1.32 | 6.11±1.37 | 3.52±1.16 | 6.08 |
| CLIP-clip (G) | 2.60±0.48 | 1.78±0.41 | 0.91±0.38 | 1.76 | 7.19±1.10 | 10.94±1.30 | 5.47±1.02 | 7.87 |
| CLIP-clip (R) | 1.50±0.18 | **0.41±0.13** | **0.19±0.11** | 0.70 | **4.46±1.19** | 6.29±1.31 | 2.72±1.09 | 4.49 |
| SFID (G) | 1.09±0.18 | 0.60±0.18 | 0.42±0.14 | 0.70 | 8.07±1.28 | 7.77±1.43 | 1.37±1.04 | 5.74 |
| SFID (R) | 1.08±0.18 | 0.61±0.18 | 0.42±0.14 | 0.70 | 8.17±1.26 | 7.26±1.47 | 1.94±1.09 | 5.79 |
| DeAR | 1.33±0.19 | 0.59±0.16 | 0.36±0.13 | 0.76 | 7.98±1.30 | 5.59±1.29 | 3.52±1.15 | 5.70 |
| VDD | 5.34±0.64 | 1.52±0.49 | 0.58±0.38 | 2.48 | 7.87±1.21 | 6.19±1.29 | **1.02±0.75** | 5.03 |
| Prompt | 1.05±0.22 | 0.83±0.21 | 0.39±0.18 | 0.76 | 8.41±1.25 | 5.01±1.24 | 2.73±1.19 | 5.38 |
| **ALA-BA** | 1.04±0.17 | 0.59±0.16 | 0.33±0.14 | 0.65 | 6.50±1.34 | **3.70±1.11** | 3.23±1.19 | 4.48 |
| **ALA-N** | **0.91±0.15** | 0.62±0.16 | 0.27±0.13 | **0.60** | 4.64±0.73 | 4.31±0.77 | 2.49±0.61 | **3.81** |

Table 8: Experimental results for the VQA-as-judge task on the FACET dataset. Red indicates notable degradation. ALA-BA preserves the original model's accuracy, showing no observed degradation, whereas other methods often reduce accuracy level.

| VQA-Task-3 Accuracy ($\uparrow$) | LLaVA-1.5 | | | | PaliGemma | | | |
|---|---|---|---|---|---|---|---|---|
| | Female | Male | Overall | $D_{WCA}$ | Female | Male | Overall | $D_{WCA}$ |
| Baseline | 88.76±0.48 | 86.34±0.32 | 86.96±0.28 | - | 82.07±0.62 | 86.45±0.33 | 85.32±0.28 | - |
| CLIP-clip | 89.07±0.50 | 85.97±0.32 | 86.77±0.28 | -0.37 | 79.47±0.63 | 88.22±0.31 | 85.96±0.27 | -2.60 |
| SFID-BA | 88.70±0.49 | 86.34±0.31 | 86.95±0.25 | -0.06 | 82.60±0.59 | 85.83±0.34 | 85.00±0.28 | -0.62 |
| DeAR | 86.53±0.54 | 87.98±0.30 | 87.60±0.26 | -2.23 | 81.60±0.59 | 86.68±0.33 | 85.36±0.28 | -0.47 |
| VDD | 88.38±0.49 | 87.01±0.31 | 87.36±0.26 | -0.38 | 81.61±0.64 | 87.01±0.32 | 85.61±0.30 | -0.46 |
| **ALA-BA** | 88.72±0.48 | 86.34±0.32 | 86.97±0.26 | **-0.04** | 82.07±0.58 | 86.41±0.32 | 85.31±0.28 | **-0.04** |

Table 9: Compute Resources Used for Experiments

| Component | Details |
|---|---|
| CPU | AMD EPYC 7313 16-Core Processor |
| GPU | NVIDIA RTX A5000 |

indicates confident steering of the language model. Its high standard deviation (1.9425) suggests this steering is adaptive, applying corrective force variably as needed.

Table 10: Overall logit entropy statistics for each debiasing method. A lower mean entropy indicates a more confident output distribution. ALA-BA's high standard deviation suggests an adaptive intervention strategy.

| Method | Mean | **Std** | Min | Max |
|---|---|---|---|---|
| Baseline | 1.7498 | 1.8109 | 0.0108 | 5.2951 |
| CLIP-clip | 1.3159 | 1.3838 | 0.0078 | 5.1445 |
| SFID | 2.2589 | 1.9786 | 0.1394 | 5.5039 |
| DeAR | 1.7498 | 1.8109 | 0.0108 | 5.2951 |
| VDD | 2.0867 | 1.8935 | 0.1099 | 5.2953 |
| Prompt | 1.8136 | 1.8240 | 0.0108 | 5.3562 |
| **ALA-BA** | 1.6713 | 1.9425 | 0.0796 | 5.2951 |

Table 11 details the nature of this intervention. The ideal debiasing method should produce a diverse, high-entropy distribution over undesirable tokens (showing no single toxic word is favored) and a focused, low-entropy distribution over desirable tokens (to show confident guidance).

- **Distribution over Undesirable Tokens:** ALA-BA achieves an entropy of 0.1925 over the toxic (High Bias) tokens. This is a high-entropy distribution compared to the baseline (0.1398), demonstrating that our method effectively diffuses focus away from any single toxic word, making the model's undesirable choices less predictable and more evenly suppressed.

- **Distribution over Desirable Tokens:** ALA-BA's intervention results in a more focused, lower-entropy distribution (0.9808) over the desirable (Low Bias) tokens compared to the baseline (1.0351). This shows that our method is not merely suppressing undesirable words, but is also decisively guiding the model's output towards a specific, high-quality set of non-toxic alternatives.

In summary, the entropy analysis demonstrates that ALA-BA successfully achieves the desired dual outcomes. It creates a diverse, high-entropy distribution over undesirable words (effective suppression) while simultaneously creating a focused, low-entropy distribution over desirable words (confident guidance), a combination not achieved by the other methods except CLIP-clip.

Table 11: Conditional entropy for undesirable/toxic (High Bias) and desirable/non-toxic (Low Bias) token sets. ALA-BA achieves the desired high-entropy distribution over toxic words and low-entropy distribution over desirable words.

| Method | High Bias Tokens Entropy | Low Bias Tokens Entropy |
|---|---|---|
| Baseline | 0.1398 | 1.0351 |
| CLIP-clip | 0.2351 | 0.7874 |
| SFID | 0.1921 | 1.2649 |
| DeAR | 0.1398 | 1.0351 |
| VDD | 0.1957 | 1.1885 |
| Prompt | 0.1461 | 1.0717 |
| **ALA-BA** | 0.1925 | 0.9808 |

## L  SCALABILITY TO SOTA MODELS: PROMPT BASELINE & INSTRUCTION FOLLOWING

We hypothesize that the failure of the "Prompt" baseline in our initial experiments might have been due to the weaker instruction-following capabilities of older models. To test this, we conducted new experiments with **Qwen2.5-VL-3B-Instruct** Team (2025), which possesses superior instruction-following capabilities.

As shown in Table 12, the "Prompt" baseline still failed, degrading fairness ($MR_C$ increased from $2.11 \rightarrow 3.56$) even on this state-of-the-art model. This confirms that the issue is not the model's ability to follow instructions, but the inherent instability of the prompting method itself.

Table 12: Results on VQA-Task-1 for Qwen2.5-VL-3B-Instruct.

| Method | $MR_{\mathcal{M}} - MR_{\mathcal{F}}$ | $MR_{\mathcal{O}}$ (Overall Error) | $\mathbf{MR}_{\mathcal{C}}$ (Composite) |
|---|---|---|---|
| **Baseline** | $0.39 \pm 0.38$ | $2.08 \pm 0.29$ | $2.11 \pm 0.29$ |
| **Prompt** | $0.89 \pm 0.80$ | $3.43 \pm 0.36$ | $3.56 \pm 0.43$ |
| **ALA-BA** | $0.42 \pm 0.41$ | $1.91 \pm 0.28$ | $1.94 \pm 0.30$ |
| **ALA-N** | $\mathbf{0.02 \pm 0.02}$ | $\mathbf{0.01 \pm 0.01}$ | $\mathbf{0.03 \pm 0.03}$ |

We attribute the failure of prompting to its inability to handle classifier uncertainty, a problem ALA is specifically designed to solve.

- **Prompting (Binary/Hard):** The prompt-based method forces a binary decision. Even if the external classifier is uncertain (e.g., 51% confidence) or incorrect on an ambiguous image, the prompt converts this into a "hard" fact: *"The gender is male."* The LMM, being a good instruction follower, adheres to this (potentially wrong) fact, leading to hallucinations or errors.

- **ALA (Continuous/Soft):** In contrast, ALA utilizes the continuous score ($s$) from the classifier. As detailed in Appendix G, our classifier yields a score near zero for ambiguous images. In these cases, ALA naturally produces a "neutralization effect" ($s \approx 0$), rather than forcing a specific gender.

In short, ALA robustly handles ambiguous inputs by neutralizing them, whereas prompting forces the model to commit to a potentially incorrect binary label.

## M    EXTENSION TO DIVERSE VQA TASKS (DISCRIMINATIVE REASONING)

Evaluating the model's capabilities beyond captioning is vital to demonstrate generality. To address this, we introduce **VQA-Task-4: Occupation Recognition**. Unlike the generative tasks (Task 1 & 2), this is a discriminative classification task that tests the model's ability to perform visual reasoning without relying on social stereotypes.

We utilize the FACET dataset to identify "confusion pairs" of occupations that are known to be heavily gender-skewed. We specifically focus on the "Doctor vs. Nurse" pair, where models frequently misclassify Female Doctors as "Nurses" due to linguistic priors. We use the discriminative prompt: *"Is this person a doctor or a nurse?"*

To solve this, we apply ALA-BA by training a lightweight profession classifier (linear layer on frozen embeddings) to serve as the external image classifier $f^{\text{image}}$.

- **Occupation Target Signal ($s$):** The image classifier analyzes the visual features (e.g., stethoscope, white coat) and outputs a target score $s$ corresponding to the true profession (e.g., "Doctor"), regardless of the subject's gender.
- **Logic:** When the LMM attempts to generate "Nurse" due to language bias (e.g., "She" $\rightarrow$ "Nurse"), ALA detects the discrepancy between the text bias $\alpha$ ("Nurse") and the image signal $s$ ("Doctor"). It then steers the logits to align with the visual evidence $s$, effectively correcting the stereotypical error.

We report the **Stereotypical Error Rate (SER)**, defined as the average of the error rates for counter-stereotypical examples: $SER = \frac{1}{2}(E_{\text{Female}\rightarrow\text{Nurse}} + E_{\text{Male}\rightarrow\text{Doctor}})$.

Table 13: Performance on VQA-Task-4 (Occupation Recognition) using LLaVA.

| Method | Female Dr $\rightarrow$ Nurse | Male Nurse $\rightarrow$ Dr | **SER** | Overall Error |
|---|---|---|---|---|
| **Baseline** | 94.43% | 8.45% | **51.44%** | 36.01% |
| **ALA** ($\lambda = 1$) | 71.78% | 4.73% | **38.26%** | 23.28% |
| **ALA** ($\lambda = 2$) | 20.91% | 7.09% | **14.00%** | **10.62%** |

The baseline results in Table 13 confirm a severe bias: LLaVA misclassifies 94.43% of Female Doctors as Nurses. Applying ALA ($\lambda = 2$) drastically reduces this stereotypical error to 20.91%, decreasing the SER to 14.00%. This result is crucial because it demonstrates that ALA is not limited to "tagging". It effectively intervenes in the model's reasoning process. By suppressing logit-level gender priors, ALA restores the model's ability to perform accurate, discriminative visual recognition.

## N    GRANULAR ERROR ANALYSIS

To provide a more transparent picture of ALA's behavior, we analyzed the absolute misclassification counts (Male-to-Female vs. Female-to-Male errors) at varying logit adjustment strengths ($\lambda$) using the PaliGemma model on VQA-Task-1.

As shown in Table 14, the baseline model is highly imbalanced, exhibiting a strong intrinsic bias toward "Female" predictions (411 M2F errors vs. 284 F2M errors). As $\lambda$ increases, the total misgendered count drops consistently. However, we observe asymmetric correction dynamics. At moderate $\lambda$ values (e.g., $\lambda = 10.0$), F2M errors are corrected aggressively (dropping by $\sim 86\%$), while M2F errors remain persistent. Because the baseline model has a dominant female prior, the logits for female tokens are highly confident. Consequently, ALA-N requires a stronger suppression force (higher $\lambda$) to override the dominant female prediction compared to the weaker male prediction. At high $\lambda$ values (e.g., $\lambda = 100.0$), this asymmetry disappears, as the suppression penalty overcomes the baseline priors entirely, forcing the model to neutral tokens and leading to a massive collapse in both error types.

Table 14: Absolute Misclassification Counts on PaliGemma (VQA-Task-1) under varying $\lambda$.

| $\lambda$ | M2F Errors | F2M Errors | Total Misgendered |
|---|---|---|---|
| 0.0 (Baseline) | 411 | 284 | 695 |
| 10.0 | 402 | 39 | 441 |
| 100.0 | 16 | 14 | 30 |

## O    SCALING TO MULTIPLE ATTRIBUTES (INTERSECTIONAL DEBIASING)

Real-world debiasing often requires addressing multiple protected attributes simultaneously (e.g., Race and Gender). To address this, we extended our framework to an **Intersectional Logit Processor**. A key advantage of ALA is its **signal agnosticism**: it can ingest debiasing signals from diverse sources (neural or symbolic) and aggregate them into a single logit adjustment.

We sum the adjustments from independent sources:

$$\mathbf{z}' = \mathbf{z} - \lambda_{\text{gender}}(\alpha_{\text{gender}} - s_{\text{gender}})\beta_{\text{gender}} - \lambda_{\text{race}}(\alpha_{\text{race}} - s_{\text{race}})\beta_{\text{race}} \quad (7)$$

We demonstrate the adaptability of ALA to data availability by integrating two different types of signals:

1. **Gender (Trained):** We use our standard trained classifier ($f_{\text{gender}}^{\text{text}}$) because high-quality labeled data (Bias-in-Bios) is available to learn complex gender signals.

2. **Race (Rule-Based):** For race, no comparable large-scale dataset exists to train a reliable text classifier for short captions. To overcome this data scarcity, we implemented a rule-based symbolic detector. This detector identifies the presence of explicit racial descriptors (e.g., "Indian", "Asian") via dictionary matching. We then set the target $s_{\text{race}}$ to eliminate this signal.

We applied this joint adjustment to the FACET dataset and evaluated it on three specific metrics to quantify intersectional fairness:

- **Avg Racial:** The average frequency of explicit racial descriptors appearing in the generated text across all evaluated images.

- **Max Gap:** The maximum disparity in the frequency of racial descriptors between any two distinct racial groups, measuring parity across demographics.

- **Gender $\text{MR}_C$:** The composite gender misclassification rate, as defined in Section 3.1.

Table 15: Performance of Intersectional Debiasing on FACET using LLaVA.

| Method | Avg Racial ($\downarrow$) | Max Gap ($\downarrow$) | Gender $MR_C$ ($\downarrow$) |
|---|---|---|---|
| **Baseline** | $28.21 \pm 6.67$ | $6.77 \pm 0.82$ | $6.89 \pm 0.72$ |
| **Gender Debias Only** | $31.63 \pm 0.70$ | $6.54 \pm 1.83$ | $\mathbf{3.80 \pm 0.55}$ |
| **Race Debias Only** | $\mathbf{0.19 \pm 0.08}$ | $\mathbf{0.29 \pm 0.21}$ | $6.92 \pm 0.73$ |
| **Gender + Race (Intersectional)** | $\underline{0.71 \pm 0.11}$ | $\underline{0.65 \pm 0.21}$ | $\underline{3.91 \pm 0.60}$ |

The results in Table 15 confirm the scalability of our method. Applying Gender Debias Only effectively reduces gender bias ($MR_C$ $6.89 \rightarrow 3.80$) but leaves racial descriptors high. Conversely, Race Debias Only removes racial keywords (Avg Racial $28.21 \rightarrow 0.19$) but neglects gender. The intersectional configuration successfully combines both, reducing Avg Racial to 0.71 and Gender $MR_C$ to 3.91 simultaneously without conflict.

The following captions generated for a single image illustrate the specific effect of each configuration:

- **Baseline:** "The image features a **woman** dressed in traditional **Indian** garb, standing on a stage and performing a dance. She is holding a large, colorful bowl on her head..."

- **Gender Debias:** "In a lively outdoor scene, a group of people is enjoying a festival. A beautiful **Indian dancer** is performing a dance, holding a large orange bowl..." *(Removes gendered "woman", preserves racial "Indian")*

- **Race Debias:** "The image features a **woman** dressed in traditional clothing, standing on a stage and performing a dance. She is holding a large, colorful bowl on her head..." *(Removes racial "Indian", preserves gendered "woman")*

- **Intersectional Debias:** "In a lively outdoor scene, a group of people is enjoying a festival. A beautiful young **dancer** is performing a traditional dance, standing on a platform and holding a large bowl on top of it." *(Removes both attributes)*

## P   THE USE OF LARGE LANGUAGE MODELS (LLMs)

We employed an LLM as a writing assistant during the preparation of this manuscript. The LLM's role was strictly limited to improving the grammar, clarity, and phrasing of existing text. Each modification suggested by the model was carefully reviewed by the authors to ensure the original scientific meaning and technical details remained accurate and unchanged. The LLM did not contribute to the core research ideation, methodology, experimental results, or conclusions presented in this paper. The authors take full responsibility for all content in this manuscript.

