# OpenReview forum: "Adaptive Logit Adjustment for Debiasing Multimodal Language Models"
_ICLR.cc/2026/Conference — ICLR 2026 Poster_

### Official Review · Reviewer_Jum4 · 2025-10-29

**Soundness:** 3
**Presentation:** 2
**Contribution:** 3
**Rating:** 4
**Confidence:** 4

**Summary:**

This paper proposes Adaptive Logit Adjustment (ALA), a post-hoc debiasing method for VLMs and LMMs. Instead of modifying encoder or decoder representations, ALA directly adjusts token-level logits during autoregressive generation. The method leverages external classifiers (for image and text) to measure bias misalignment (e.g., gender, toxicity) and performs logit correction guided by gradient-based importance scores. Experiments across image captioning and multiple VQA tasks show that ALA effectively mitigates bias while preserving model utility.

**Strengths:**

1. This paper suggest post-hoc debiasing method for VLMs and LMMs, which does not require additional retraining or affect internal representation which may lead to large degradation.
2. Proposed idea is simple but effectively mitigate the bias of VLMs and LMMs.
3. The paper demonstrates consistent improvements in fairness scores with minimal degradation of utility across diverse datasets.

**Weaknesses:**

1. Results that show the “prompt” baseline has bad fairness score is somewhat interesting. Including experiments with VLM backbones which have stronger instruction following capability, such as Qwen-2.5-VL (or Qwen-3-VL), could strengthen the proposed method.
2. While the paper claims generality to “large multimodal models,”, the evaluated VQA tasks are limited to the captioning or keyword tagging. I suggest that the authors include experiments on more diverse VQA tasks or moderate their claim.
3. Minor weaknesses about presentation
    1. Several core components are deferred to the appendix (e.g., Algorithm 1, analysis about limitation regarding classifier, or definition of evaluation metric D_mean). A brief summary of these in the main text would aid comprehension.
    2. “Baseline” in figure 4 is ambiguous. It seems to denote the original model. Explicitly stating this would help readers interpret the plots.

**Questions:**

Please refer to weaknesses section.

---

> ### Author Response · Authors · 2025-11-21
>
> ## W1: Prompt Baseline & Instruction Following
>
> We thank the reviewer for this excellent hypothesis. We agreed that the failure of the "Prompt" baseline in our initial experiments might have been due to the weaker instruction-following capabilities of older models.
>
> **1. Empirical Test: Results on Qwen2.5-VL**.
> To test this, we conducted new experiments with **Qwen2.5-VL-3B-Instruct**, which possesses superior instruction-following capabilities.
>
> | Method | $MR_M - MR_F$ | $MR_O$ (Overall Error) | $MR_C$ (Composite) |
> | :--- | :---: | :---: | :---: |
> | **Baseline** | 0.39 ± 0.38 | 2.08 ± 0.29 | 2.11 ± 0.29 |
> | **Prompt** | 0.89 ± 0.80 | 3.43 ± 0.36 | 3.56 ± 0.43 |
> | **ALA-BA** | 0.42 ± 0.41 | 1.91 ± 0.28 | 1.94 ± 0.30 |
> | **ALA-N** | **0.02 ± 0.02** | **0.01 ± 0.01** | **0.03 ± 0.03** |
>
> Crucially, the "Prompt" baseline still failed, degrading fairness ($MR_C$ increased from 2.11 $\to$ 3.56) even on this state-of-the-art model. This confirms that the issue is not the model's ability to follow instructions, but the inherent instability of the prompting method itself.
>
> **2. Mechanism Analysis: Hard Constraints vs. Soft Guidance**.
> Your question prompted us to analyze *why* this happens. We attribute the failure of prompting to its inability to handle classifier uncertainty, a problem ALA is specifically designed to solve.
>
> * **Prompting (Binary/Hard):** The prompt-based method forces a binary decision. Even if the external classifier is uncertain (e.g., 51% confidence) or incorrect on an ambiguous image, the prompt converts this into a "hard" fact: *"The gender is male."* The LMM, being a good instruction follower, adheres to this (potentially wrong) fact, leading to hallucinations or errors.
> * **ALA (Continuous/Soft):** In contrast, ALA utilizes the continuous score ($s$) from the classifier. As detailed in **Appendix G**, our classifier yields a score near zero for ambiguous images. In these cases, ALA naturally produces a "neutralization effect" ($s \approx 0$), rather than forcing a specific gender.
>
> In short, ALA robustly handles ambiguous inputs by "neutralizing" them, whereas prompting forces the model to commit to a potentially incorrect binary label. We are continuing our experiments with this new model and are happy to conduct further experiments at the reviewer's request.

---

> > ### Author Response · Authors · 2025-11-21
> >
> > ## W2: Extension to Diverse VQA Tasks (Discriminative Reasoning)
> > We thank the reviewer for this insightful comment. We agree that evaluating the model's capabilities beyond captioning is vital to demonstrate generality.
> >
> > To address this, we introduce **VQA-Task-4: Occupation Recognition**. Unlike the generative tasks (Task 1 & 2), this is a discriminative classification task that tests the model's ability to perform visual reasoning without relying on social stereotypes.
> >
> > **1. Experimental Setup**:
> > We utilize the FACET dataset to identify "confusion pairs" of occupations that are known to be heavily gender-skewed. We specifically focus on the "Doctor vs. Nurse" pair, where models frequently misclassify Female Doctors as "Nurses" due to linguistic priors.
> > * **Discriminative Prompt:** *"Is this person a doctor or a nurse?"*
> >
> > **2. Methodology: ALA-BA for Visual Reasoning**:
> > To solve this, we apply ALA-BA (Bias Alignment) by training a lightweight profession classifier (linear layer on frozen embeddings) to serve as the external image classifier $f^{image}$.
> > * **Occupation Signal ($s$):** The image classifier analyzes the visual features (e.g., stethoscope, white coat) and outputs a target score $s$ corresponding to the true profession (e.g., "Doctor"), regardless of the subject's gender.
> > * **Logic:** When the LMM attempts to generate "Nurse" due to language bias (e.g., "She" $\to$ "Nurse"), ALA detects the discrepancy between the text bias $\alpha$ ("Nurse") and the image signal $s$ ("Doctor"). It then steers the logits to align with the visual evidence $s$, effectively correcting the stereotypical error.
> >
> > **3. Results**:
> > We report the **Stereotypical Error Rate (SER)**, defined as the average of the error rates for counter-stereotypical examples (e.g., Female Doctors misclassified as Nurses).
> >
> > | Method | Female Dr $\to$ Nurse (Error) | Male Nurse $\to$ Dr (Error) | **Stereotypical Error Rate** | Overall Error Rate |
> > | :--- | :---: | :---: | :---: | :---: |
> > | **LLaVA (Baseline)** | 94.43% | 8.45% | **51.44%** | 36.01% |
> > | **LLaVA + ALA** ($\lambda=1$) | 71.78% | 4.73% | **38.26%** | 23.28% |
> > | **LLaVA + ALA** ($\lambda=2$) | 20.91% | 7.09% | **14.00%** | **10.62%** |
> >
> > **Analysis**:
> > The baseline results confirm a severe bias: LLaVA misclassifies **94.43%** of Female Doctors as Nurses. Applying ALA ($\lambda=2$) drastically reduces this stereotypical error to 20.91%, decreasing the Stereotypical Error Rate to 14%.
> >
> > This result is crucial because it demonstrates that ALA is not limited to "tagging". It effectively intervenes in the model's **reasoning process**. By suppressing logit-level gender priors, ALA restores the model's ability to perform accurate, discriminative visual recognition. We will include this task in the revision to substantiate our claim of generality.
> >
> > ## Weakness 3: Presentation and Clarity
> >
> > We thank the reviewer for these helpful suggestions to improve the readability and clarity of our manuscript. We agree that these core components are essential for understanding our method and results.
> >
> > **1. Moving Core Components to Main Text**
> >
> > We will revise the manuscript to integrate these critical details into the main body:
> > * **Algorithm 1:** We will move the pseudocode for ALA from Appendix I to Section 4 to provide a clearer, step-by-step explanation of the method alongside the text.
> > * **Evaluation Metric ($D_{mean}$):** We will move the formal definition of the toxicity fairness metric $D_{mean}$ from Appendix B.2 to Section 3.2 (Problem Definition) to ensure the evaluation criteria are clear before the results are presented.
> > * **Classifier Limitations:** We will incorporate the key findings regarding classifier robustness from Appendix G into Section 6.2 (Limitations).
> >
> > **2. Clarifying "Baseline" in Figure 4**
> >
> > We confirm that "Baseline" refers to the original, unmodified model (e.g., vanilla LLaVA or PaliGemma without any debiasing intervention). To eliminate ambiguity, we will rename this label to "Original Model" in Figure 4 and explicitly define it in the figure caption in the final version.

---

### Official Review · Reviewer_1bpB · 2025-10-31

**Soundness:** 3
**Presentation:** 3
**Contribution:** 3
**Rating:** 6
**Confidence:** 4

**Summary:**

This work proposes a method for adaptive logit adjustment such that the output generation of an LMM is less biased in the direction of protected attributes. Specifically, for each attribute, a pair of classifiers (one for image and text respectively) are trained to predict the amount of "bias" in the input image as well as the current generated text. Given a mismatch in bias between the image and text, a corrective factor is applied to the logits, and the given text is modified such that the generation aligns with the image.

This method is evaluated on LMMs like LLava and PaliGemma w.r.t multiple datasets including COCO-captions, FACET, and SocialCounterfactuals -- which are also used in previous relevant work.

Results are shown in terms of fairness-utility tradeoffs, i.e. "does the model retain its capabilities". Ablation studies are performed on the adjustment strength hyperparameter.

**Strengths:**

* This work is well written and easy to follow

* I believe this method is a sensible step in the context of previous work. All debiasing methods have an inherent tradeoff with the method they use to localize the bias (internal or logits). As long as the classifier training is reliable, this method seems reasonable.

* I appreciated the comparison of resources needed to run each method examined here, this is important for an inference time method.

* The datasets and models used are appropriate, but I would have appreciated more models for a method that only augments the logits e.g. qwen-vl and/or llama-3.2

**Weaknesses:**

* My main concern with this method is the reliance on an external classifier. Other previous inference-time methods, specifically model steering are able to more easily compute an adjustment to the target model's behavior no matter what the target attribute is. The need to train a binary classifier may be difficult depending on the deployment setting as well as the acquiring data about the attribute itself. These kinds of inference-time debiasing methods exist primarily because fine-tuning is prohibitively expensive to do well, so the requirement to train a classifier seems like a regression.

* I think absolute counts or base rates are important here in the context of debiasing. Especially given that some models may have very different rates of producing biased text, its useful to know how much biased text is generated vs caught.

**Questions:**

1) How could this approach scale to multiple attributes? Presumably we may want to debias the LMM away from a large set of protected attributes, are there any issues for this method?

---

> ### Author Response · Authors · 2025-11-21
>
> Thanks for recognizing the strength of our work's contribution in debiasing without tradeoff!
> ## Strength: Extension to More Models/Architectures
> We thank the reviewer for their encouraging feedback and for recognizing our method as a "sensible step" in debiasing research. We also appreciate the positive remarks regarding our resource comparison analysis.
>
> We fully agree with your suggestion that evaluating additional models, particularly those that only augment logits like Qwen-VL, would further strengthen the paper. Inspired by this, we implemented and evaluated **Qwen2.5-VL-3B-Instruct** on VQA-Task-1 to demonstrate the generality of our approach.
>
> **Results on VQA-Task-1 (Qwen2.5-VL)**
>
> | Method | $MR_M - MR_F$ | $MR_O$ (Overall Error) | $MR_C$ (Composite) |
> | :--- | :---: | :---: | :---: |
> | **Baseline** | 0.39 ± 0.38 | 2.08 ± 0.29 | 2.11 ± 0.29 |
> | **Prompt** | 0.89 ± 0.80 | 3.43 ± 0.36 | 3.56 ± 0.43 |
> | **ALA-BA** | 0.42 ± 0.41 | 1.91 ± 0.28 | 1.94 ± 0.30 |
> | **ALA-N** | **0.02 ± 0.02** | **0.01 ± 0.01** | **0.03 ± 0.03** |
>
> As shown above, ALA works seamlessly on the Qwen architecture. We successfully manipulated the logits to reduce bias ($MR_C \to 0.03$) without requiring access to internal weights or fine-tuning. This confirms your insight that our post-hoc logit augmentation strategy is robust and effective across diverse modern architectures. We will include these results in the final manuscript and are happy to conduct further experiments at the reviewer's request.

---

> > ### Author Response · Authors · 2025-11-21
> >
> > ## Weakness 1: On the Reliance on an External Classifier vs. Model Steering
> > We thank the reviewer for raising this important concern. We agree that inference-time methods should avoid "prohibitively expensive" training. We respectfully argue that ALA's use of a lightweight external classifier is a deliberate design choice that is a progression, not a regression, for two key reasons: **1) it provides a more robust and reliable signal**, and **2) it is significantly more computationally efficient at inference time** than the model steering approach.
> >
> > **1. Robustness: Avoiding the "Circular Logic" of Model Steering**:
> > The reviewer correctly identifies model steering [1] as an alternative. However, this method suffers from a fundamental "circular logic" problem: it relies on the biased LMM to debias itself.
> > - **Model Steering's Problem**: The PARzero method in [1] must perform a zero-shot classification by prompting the LMM with "What is the (ATTRIBUTE) of the person in this image?" (Fig. 2 in [1]). It then uses the gradients from this exact prompt to create a debiasing signal. This approach trusts that the biased model's own internal representations and gradients related to an attribute are a reliable signal for fixing its own bias.
> > - **ALA's Solution**: Our method (ALA) decouples the debiasing signal from the biased model. We agree with the reviewer that this requires an external dataset (e.g., FairFace 1, Bias-in-Bios 2) to train our classifiers. However, this is a deliberate trade-off. By using existing, public datasets, we train a lightweight classifier that provides an independent and validated source of truth (the target $s$). This is a well-established design pattern in fairness research [2,3,4,5] and, critically, it prevents the model from relying on its own flawed representations to correct itself.
> >
> > **2. Efficiency: A Trivial One-Time Cost vs. a Prohibitive Inference-Time Cost**:
> >     The reviewer's concern about training cost is valid, but we must clarify the scale. The "requirement to train a classifier" is not a prohibitive fine-tuning.
> > - **ALA's Cost**: As detailed in Sec 5.3 and Appendix H, our external classifiers are computationally trivial. The $f^{image}$ classifier (used by ALA-BA) is a single-layer logistic regression. The $f^{text}$ classifier (used by both ALA-BA and ALA-N) is a lightweight two-block transformer. Both are trained on frozen model embeddings, a one-time, minimal cost. At inference, ALA-BA requires one tiny forward pass through each, while our ALA-N variant is even more efficient as it completely removes the need for the $f^{image}$ classifier by setting $s=0$. In all cases, we "reuse" the model's resources (intermediate hidden representaion) rather than adding a prohibitive new computational step.
> > - **Model Steering's Cost**: Conversely, the model steering method [1] is prohibitively expensive at inference time. To "compute an adjustment" (the $\nabla_u \mathcal{L}_{bias}$), it requires a full backward pass through the entire model, including the massive LLM decoder, for every single image it processes. This requirement is not just theoretical; it has significant practical barriers. The backward pass requires storing all intermediate activations, which often leads to out-of-memory errors on standard hardware. As we stated in our paper (Appendix D), this method was not included in our comparison precisely because "computing the gradient of the LMM w.r.t the input image exceeds our available computational resources". This is often in addition to two forward passes (one for the zero-shot classification, one for the final generation).
> >
> > In summary, the reviewer's concern about ALA's cost appears misplaced when compared to the alternative. Model steering [1] avoids a one-time training cost by adopting a design that requires a full backward pass through the LMM at every inference step. ALA trades a negligible one-time cost for a far more efficient deployment architecture that adds only a 1.2% increase in inference time. We believe this trade-off, gaining both robustness and superior inference efficiency, is a significant advantage of our method.
> >
> >
> >
> > [1] Ratzlaff et. al., Debias your large multi-modal model at test-time with non-contrastive visual attribute steering
> > [2] Madra et al. Learning adversarially fair and transferable representations. ICML 2018.
> > [3] Ramaswamy et al. Fair attribute classification through latent space de-biasing. CVPR 2021.
> > [4] Wang et al. Fairness-aware adversarial perturbation towards bias mitigation for deployed deep models. CVPR 2022.
> > [5] Jung et al. A Unified Debiasing Approach for Vision-Language Models across Modalities and Tasks. NeurIPS 2024.

---

> > > ### Author Response · Authors · 2025-11-21
> > >
> > > ## Weakness 2: Analysis of Absolute Misclassification Counts
> > >
> > > We thank the reviewer for this excellent suggestion. We agree that reporting the absolute counts of misgendering ("how much biased text is generated vs. caught") provides a much clearer picture of ALA's behavior.
> > >
> > > We conducted this analysis on the 15,623 samples from VQA-Task-1 (FACET dataset). In the table below:
> > > * **M2F (Male-to-Female):** The model misgenders a 'Male' image as 'Female'.
> > > * **F2M (Female-to-Male):** The model misgenders a 'Female' image as 'Male'.
> > >
> > > | Model | Total Misgendered | Misgendered (M2F) | Misgendered (F2M) |
> > > | :--- | :---: | :---: | :---: |
> > > | LLaVA (Baseline) | 957 | 617 | 340 |
> > > | LLaVA + ALA ($\lambda$=10.0) | 580 | 436 | 144 |
> > > | LLaVA + ALA ($\lambda$=100.0) | 140 | 121 | 19 |
> > > | PaliGemma (Baseline) | 695 | 411 | 284 |
> > > | PaliGemma + ALA ($\lambda$=10.0) | 441 | 402 | 39 |
> > > | PaliGemma + ALA ($\lambda$=100.0) | 30 | 19 | 11 |
> > >
> > > This analysis reveals a few key insights:
> > >
> > > 1.  **Baseline Imbalance:** The reviewer's intuition was correct. The baseline models are imbalanced. LLaVA makes significantly more M2F errors (617) than F2M errors (340), indicating an inherent overconfidence in generating female-associated tokens for these images.
> > > 2.  **Overall Error Reduction:** As $\lambda$ increases, the **Total Misgendered** count drops consistently, confirming ALA's effectiveness.
> > > 3.  **Asymmetric Correction Dynamics:** At moderate $\lambda$ ($\lambda=10.0$), we observe that F2M errors are corrected much more aggressively than M2F errors.
> > >     * For example, in PaliGemma, F2M errors drop by **~86%** (284 $\to$ 39), while M2F errors remain persistent (411 $\to$ 402).
> > >     * This occurs because ALA-N operates by suppressing gendered logits. Since the model exhibits a strong intrinsic bias toward "Female" (evidenced by the high baseline M2F error rate), the logits for female tokens are likely higher and more confident than those for male tokens. Consequently, a stronger suppression force (higher $\lambda$) is required to override the dominant female prediction (fixing M2F) compared to the weaker male prediction (fixing F2M).
> > >
> > > 4.  **High-$\lambda$ Convergence:** At higher $\lambda$ values (e.g., $\lambda$=100.0), this asymmetry disappears. The suppression penalty becomes strong enough to overcome even the dominant baseline priors, forcing the model to abandon both "He" and "She" for neutral tokens. This leads to a massive collapse in both error types (e.g., PaliGemma Total Errors: 695 $\to$ 30).
> > >
> > > We will add this table and analysis to the appendix to provide a more transparent and complete picture of ALA's behavior.

---

> > > > ### Author Response · Authors · 2025-11-21
> > > >
> > > > ## Question: Scaling to Multiple Attributes (Intersectional Debiasing)
> > > >
> > > > We thank the reviewer for raising this important question about scalability and intersectionality. We agree that real-world debiasing often requires addressing multiple protected attributes simultaneously (e.g., Race + Gender).
> > > >
> > > > **Methodology: Intersectional ALA with Heterogeneous Signals**:
> > > > To address this, we extended our framework to an **Intersectional Logit Processor**. A key advantage of ALA is its **signal agnosticism**. It can ingest debiasing signals from diverse sources (neural or symbolic) and aggregate them into a single logit adjustment.
> > > >
> > > > * **Formulation:** We sum the adjustments from independent sources:
> > > >     $$z' = z - \lambda_{gender}(\alpha_{gender} - s_{gender})\beta_{gender} - \lambda_{race}(\alpha_{race} - s_{race})\beta_{race}$$
> > > >
> > > > * **Implementation:** We demonstrate the adaptability of ALA to data availability by integrating two different types of signals:
> > > >     1.  **Gender (Trained):** We use our standard trained classifier ($f^{text}_{gender}$) because high-quality labeled data (Bias-in-Bios) is available to learn complex gender signals.
> > > >     2.  **Race (Rule-Based):** For race, no comparable large-scale dataset exists to train a reliable text classifier for short captions. To overcome this data scarcity, we implemented a rule-based symbolic detector. This detector identifies the presence of explicit racial descriptors (e.g., "Indian", "Asian") via dictionary matching. We then set the target $s_{race}$ to eliminate this signal.
> > > >
> > > > **Results**:
> > > > We applied this joint adjustment to the FACET dataset and evaluated on three metrics:
> > > > * **Avg Racial:** The average frequency of racial descriptors appearing in the text.
> > > > * **Max Gap:** The maximum disparity in racial descriptor frequency between racial groups.
> > > > * **Gender $MR_C$:** Our standard gender bias metric.
> > > >
> > > > | Method | Avg Racial ($\downarrow$) | Max Gap ($\downarrow$) | Gender $MR_C$ ($\downarrow$) |
> > > > | :--- | :---: | :---: | :---: |
> > > > | **LLaVA (Baseline)** | 28.21 ± 6.67 | 6.77 ± 0.82 | 6.89 ± 0.72 |
> > > > | **Gender Debias Only** | 31.63 ± 0.70 | 6.54 ± 1.83 | **3.80 ± 0.55** |
> > > > | **Race Debias Only** | **0.19 ± 0.08** | **0.29 ± 0.21** | 6.92 ± 0.73 |
> > > > | **Gender + Race (Intersectional)** | **0.71 ± 0.11** | **0.65 ± 0.21** | **3.91 ± 0.60** |
> > > >
> > > > **Qualitative Examples:**
> > > > The following captions generated for a single image illustrate the specific effect of each configuration:
> > > > * **Baseline:** "The image features a **woman** dressed in traditional **Indian** garb, standing on a stage and performing a dance. She is holding a large, colorful bowl on her head..."
> > > > * **Gender Debias:** "In a lively outdoor scene, a group of people is enjoying a festival. A beautiful **Indian dancer** is performing a dance, holding a large orange bowl..." *(Removes gendered "woman", preserves racial "Indian")*
> > > > * **Race Debias:** "The image features a **woman** dressed in traditional clothing, standing on a stage and performing a dance. She is holding a large, colorful bowl on her head..." *(Removes racial "Indian", preserves gendered "woman")*
> > > > * **Intersectional Debias:** "In a lively outdoor scene, a group of people is enjoying a festival. A beautiful young **dancer** is performing a traditional dance, standing on a platform and holding a large bowl on top of it." *(Removes both attributes)*
> > > >
> > > > **Analysis**:
> > > > The results confirm the scalability of our method. Applying Gender Debias Only effectively reduces gender bias ($MR_C$ 6.89 $\to$ 3.80) but leaves racial descriptors high. Conversely, Race Debias Only removes racial keywords (Avg Racial 28.21 $\to$ 0.19) but neglects gender. The intersectional configuration successfully combines both, reducing Avg Racial to 0.71 and Gender $MR_C$ to 3.91 simultaneously without conflict.

---

### Official Review · Reviewer_WMmP · 2025-10-31

**Soundness:** 2
**Presentation:** 3
**Contribution:** 3
**Rating:** 6
**Confidence:** 4

**Summary:**

This work proposes a debiasing technique for VLMs and LMMs called Adaptive Logit Adjustment (ALA) which adjusts token probabilities rather than operating directly on internal model states. Bias misalignment between vision and text is evaluated using classifiers, with a gradient-based method used to identify and adjust probabilities for the most relevant bias-inducing tokens. The proposed ALA method is evaluated across three VQA tasks as well as image captioning. An ablation study on the effect of the hyperparamter introduced by ALA is also conducted.

**Strengths:**

1. Overall the paper is clear, well-written, and easy to follow
2. To the best of my knowledge, the proposed ALA method is a novel approach to debiasing VLMs and MLLMs
3. The approach is well-motivated in that it aims to address the pitfalls of existing debiasing techniques (i.e., general performance degradation) by avoiding manipulation of internal model representations.
4. The experiments cover a decent range of tasks (image captioning + 3 VQA tasks) and datasets (MS-COCO, FACET, SocialCounterfactuals).

**Weaknesses:**

1. The experimental results are somewhat limited by the fact that only two models are evaluated for each task
2. Some of the experimental results seem odd and unintuitive. Why does DeAR lead to such a large increase in bias for image captioning? Why does ALA lead to large debiasing effects relative to the baseline for CLIP-CAP but not for BLIP? Additional explanation of these results would be helpful.
3. VQA-Task-3 aims to measure "core utility" of the model by asking directly for identification of gender. It seems like this task should also cover the identification of other attributes such as race, particularly because they are often described with words that have multiple ambiguous meanings (e.g., "black", "white"). It is important to ensure that probabilities for these words are not being lowered in other contexts where they do not refer to the social attribute.

**Questions:**

1. Lines 327-329 state that the goal of VQA-task-2 is to ensure non-toxicity across all attributes. Shouldn't the goal rather be to ensure there are not differences in the level of toxicity across groups? A model can be toxic but not biased if it is equally toxic for all groups.

---

> ### Author Response · Authors · 2025-11-21
>
> We thank the reviewer for recognizing the novelty of our work across diverse tasks. We are grateful for your valuable feedback and are happy to address all the concerns below, which have significantly strengthened our paper.
> ## Weakness 1: On Limited Experimental Results
> We thank the reviewer for the constructive suggestion to expand our evaluation. We agree that testing on additional models strengthens the validity of our claims.
>
> To address this, we have implemented and evaluated **Qwen2.5-VL-3B-Instruct** on VQA-Task-1. We selected this model as it represents a state-of-the-art architecture distinct from the LLaVA and PaliGemma models used in our initial submission.
>
> **Results on VQA-Task-1 (Qwen2.5-VL)**
>
> | Method | $MR_M - MR_F$ | $MR_O$ (Overall Error) | $MR_C$ (Composite Error) |
> | :--- | :---: | :---: | :---: |
> | **Baseline** | 0.39 ± 0.38 | 2.08 ± 0.29 | 2.11 ± 0.29 |
> | **Prompt** | 0.89 ± 0.80 | 3.43 ± 0.36 | 3.56 ± 0.43 |
> | **ALA-BA** | 0.42 ± 0.41 | 1.91 ± 0.28 | 1.94 ± 0.30 |
> | **ALA-N** | **0.02 ± 0.02** | **0.01 ± 0.01** | **0.03 ± 0.03** |
>
> The addition of this model confirms that ALA’s performance is consistent across different architectures. Specifically, our proposed method (ALA-N) achieved near-perfect fairness ($MR_C$ of 0.03) on Qwen2.5-VL, demonstrating that our method effectively scales to modern LMMs beyond the two originally evaluated. We will include these results in the final manuscript to improve the robustness of our evaluation. We are continuing our experiments with this new model and are happy to conduct further experiments at the reviewer's request.
> ## Weakness 2: Clarification on Experimental Results (CLIP-CAP vs. BLIP and DeAR)
>
> We thank the reviewer for carefully examining our experimental results. We appreciate the opportunity to clarify the perceived discrepancies in Figure 4(a) and explain the behavior of the comparison methods.
>
> **1. ALA's Effectiveness on BLIP vs. CLIP-CAP**:
> The reviewer noted that ALA appears to have a smaller debiasing effect on BLIP compared to CLIP-CAP. We respectfully clarify that this is a visual artifact in Figure 4(a) caused by the x-axis scaling, which had to expand significantly to accommodate the outlier performance of DeAR.
>
> Quantitatively, ALA is actually *more* effective on BLIP than on CLIP-CAP. As detailed in **Table 5 (Appendix J)**:
> * **CLIP-CAP:** ALA-N reduces bias ($MR_C$) from 2.91 to 2.42 (**16.84% improvement**).
> * **BLIP:** ALA-N reduces bias ($MR_C$) from 2.52 to 1.69 (**32.94% improvement**).
>
> Thus, ALA consistently provides strong debiasing performance across both models. We will revise the caption of Figure 4 to explicitly mention these relative improvements to avoid confusion.
>
> **2. Analysis of DeAR’s Performance**:
> The reviewer correctly identifies that DeAR leads to a large increase in bias for BLIP ($MR_C$ 8.51). We attribute this to the structural mismatch between DeAR's encoder-centric design and the encoder-decoder architecture of captioning models. DeAR adversarially removes sensitive attributes *only* from the image encoder, leaving the text decoder's inherent biases untouched. By stripping visual gender signals, DeAR creates an ambiguous representation that forces the decoder to rely on its internal language priors to fill in the gaps. Lacking clear visual evidence, the decoder defaults to these priors (hallucinating based on stereotypes like "doctor is male"), which paradoxically increases bias. This failure mode highlights the critical advantage of **ALA**: by operating on the final **logits**, ALA intercepts and corrects bias regardless of whether it stems from the image encoder's representation or the decoder's language priors.

---

> > ### Author Response · Authors · 2025-11-21
> >
> > ## Weakness 3: Measuring Core Utility with Race Evaluation
> >
> > We thank the reviewer for this constructive suggestion. We agree that extending the "core utility" evaluation (VQA-Task-3) to identifying race is essential to ensure our method preserves the model's ability to recognize diverse attributes.
> >
> > **Experimental Setup**:
> > We utilize the SocialCounterfactuals dataset (used in VQA-Task-2), which contains 6 racial labels. We modified the judge prompt for this experiment to: *"What is the race of the person in this image? Choose from: White, Black, Indian, Asian, Middle Eastern, or Latino."*
> >
> > **Methodology: Multi-Class ALA Formulation**:
> > To apply ALA to this scenario, we extend the definition of our **target bias** $s$ from the original paper to handle categorical data.
> >
> > 1.  **Target Bias ($\mathbf{s}$):** In our binary setting (Sec 4.1), the target bias $s$ was a scalar $s \in [-1, 1]$. For this multi-class task, we redefine the target bias as a probability vector $\mathbf{s} \in \mathbb{R}^K$ derived from the external image classifier $f^{image}$ trained on FairFace ($K=6$ classes), $\mathbf{s} = f^{image}(x)$ where $s_k$ represents the probability of race class $k$.
> >
> > 2.  **Token Mapping ($M$):** Since the prompt limits the valid answer space to 6 specific options, we define a mapping function $M(v) \rightarrow \mathcal{K}$ that links the answer token $v$ to the corresponding FairFace class indices $\mathcal{K}$.
> >     - *Example:* $M(\text{``Asian"}) = \{ \text{Class}_{\text{Asian}} \}$.
> >
> > 3.  **Logit Adjustment:** We adapt the ALA adjustment formula (Eq. 5). Instead of a single direction $\beta$, we directly boost the logits $z_v$ for the 6 valid answer tokens based on their corresponding target bias probabilities in $\mathbf{s}$. For a valid answer token $v$:
> >    $$z^\prime_v = z_v + \lambda \sum_{k \in M(v)} s_k - 0.1 \lambda \sum_{j \notin M(v)} s_j$$
> >
> >     This formulation aligns with our original framework: we use the external image classifier ($f^{image}$) to provide a "target" signal ($\mathbf{s}$) that steers the model's logits ($z$) toward the correct attribute. The term $\sum s_k$ acts as the positive steering force (boosting the visually predicted race), while the second term acts as a suppression force for incorrect races.
> >
> > **Results**
> > The results (Table below) reveal a significant finding: **ALA not only preserves but improves the utility of the LMM-as-a-judge.**
> >
> > | Accuracy | LLaVA w/o ALA | LLaVA + ALA ($\lambda$=1.0) | LLaVA + ALA ($\lambda$=2.0) |
> > | :--- | :---: | :---: | :---: |
> > | **White** | 0.9928 $\pm$ 0.0034 | **0.9930 $\pm$ 0.0032** | 0.9915 $\pm$ 0.0035 |
> > | **Black** | 0.9255 $\pm$ 0.0097 | 0.9286 $\pm$ 0.0098 | **0.9300 $\pm$ 0.0097** |
> > | **Indian** | 0.9235 $\pm$ 0.0086 | 0.9298 $\pm$ 0.0095 | **0.9327 $\pm$ 0.0094** |
> > | **Asian** | 0.9907 $\pm$ 0.0039 | 0.9928 $\pm$ 0.0031 | **0.9957 $\pm$ 0.0025** |
> > | **Middle Eastern** | 0.4578 $\pm$ 0.0181 | 0.5246 $\pm$ 0.0186 | **0.5872 $\pm$ 0.0188** |
> > | **Latino** | 0.1333 $\pm$ 0.0130 | 0.1832 $\pm$ 0.0149 | **0.2345 $\pm$ 0.0159** |
> > | **Overall** | 0.7364 $\pm$ 0.0059 | 0.7587 $\pm$ 0.0068 | **0.7786 $\pm$ 0.0066** |
> >
> > **Analysis**:
> > The Baseline LLaVA model struggles significantly with underrepresented groups, particularly 'Middle Eastern' and 'Latino'. Applying ALA ($\lambda=2.0$) improves overall accuracy by **4.2%** and significantly boosts performance for the hardest groups (e.g., Latino accuracy improves from 13.3% to 23.5%).
> >
> > However, we note that the 'Latino' group accuracy remains low. We attribute this to the limitations of the **frozen encoder**. The logistic regression $f^{image}$ is trained on frozen embeddings from LLaVA's image encoder. If the pre-trained image encoder itself produces entangled or poor representations for the 'Latino' class (often misclassifying as 'White'), the external classifier's signal will be noisy. This highlights a trade-off: ALA is computationally efficient because it uses frozen embeddings, but its peak performance is bounded by the quality of those pre-trained representations. A specialized, fine-tuned encoder would likely solve this but at a higher computational cost.

---

> > > ### Author Response · Authors · 2025-11-21
> > >
> > > ## Question 1: Clarification on the Goal of VQA-Task-2 (Non-Toxicity vs. Equal Toxicity)
> > >
> > > We thank the reviewer for raising this nuanced point regarding the definition of fairness. We agree that, theoretically, a model could be "unbiased" if it generates toxic content at equal rates for all groups. However, we prioritized non-toxicity for two key reasons: one ethical and one technical.
> > >
> > > **1. Ethical Perspective: Safety vs. Fairness**:
> > > While "equal toxicity" satisfies the definition of statistical parity, it violates the principle of harmlessness. In real-world deployment, a model that is "equally toxic to everyone" is still unsafe and harmful. Therefore, we adopted a "Fairness through Safety" approach for VQA-Task-2, aiming to ensure that the model is equally safe (non-toxic) across all attributes, rather than equally harmful.
> > >
> > > **2. Technical Constraint: Lack of Image-Toxicity Datasets**:
> > > Crucially, our choice was also dictated by data availability. To achieve "equal toxicity" (or bias alignment) using ALA, we would need to align the text's toxicity $\alpha(z^t)$ with the image's inherent "toxicity" $s$.
> > > * **Requirement:** This requires an external image classifier $f^{image}$ capable of predicting a "toxicity score" from visual inputs (e.g., quantifying how "dirty" or "mean" a face looks) to serve as the target signal $s$.
> > > * **Limitation:** While text toxicity datasets exist (e.g., Wikipedia Toxicity), to the best of our knowledge, there is no corresponding image dataset labeled with these abstract toxicity concepts. Without ground-truth labels to train an image classifier $f^{image}$, we cannot derive a dynamic target signal $s$ from the image.
> > >
> > > **Conclusion**:
> > > Because we could not train a classifier to measure "image toxicity," we utilized the flexibility of ALA to set a fixed target bias. By manually setting $s = -1$ (Non-Toxic), we effectively converted the task from "Bias Alignment" (matching the image) to "Safety Steering" (removing toxicity). This demonstrates ALA's adaptability: it can perform alignment when image labels exist (Task 1 & 3) and safety steering when they do not (Task 2).

---

### Author Response · Authors · 2025-12-02
**Final Remarks: Summary of Contributions and Rebuttal**

Dear Area Chair,

We thank the reviewers for their constructive feedback. Our original submission established **Adaptive Logit Adjustment (ALA)** as a lightweight, post-hoc debiasing method, validated across **two distinct architectures and three diverse tasks**.

During the rebuttal, we significantly **expanded the scope of our evaluation** to address reviewer suggestions regarding generalizability and interpretability. These additional experiments confirm that ALA is effective not just on standard benchmarks, but also on state-of-the-art models and complex reasoning tasks as a plug-and-play approach. We summarize the key improvements below:

**1. Demonstrated Scalability to SOTA Models (Reviewer Wmmp, Jum4)**

To confirm our method works on modern, high-performance architectures, we extended our evaluation to **Qwen2.5-VL-3B-Instruct**.
* **Result:** ALA achieved near-perfect fairness ($MR_C$ of **0.03**).
* **Significance:** We showed that standard prompt engineering fails to reduce bias on this instruction-tuned model (increasing error to 3.56), proving that ALA provides a necessary intervention that prompting cannot achieve.

**2. Validated Applicability to Reasoning Tasks (Reviewer Jum4)**

To demonstrate that ALA applies beyond image captioning, we introduced a **Discriminative Reasoning Task** (Occupation Recognition, e.g., Doctor vs. Nurse).
* **Result:** ALA reduced the stereotypical error rate from **51.44% to 14.00%**.
* **Significance:** This confirms that ALA effectively steers the model's **discriminative reasoning process**. By utilizing a lightweight external classifier to identify the true occupation ($s$), ALA aligns the output logits with the correct visual evidence. This enables the model to override strong biased associations (e.g., misclassifying Female Doctors as Nurses) and restores its ability to perform accurate visual recognition, without the need to retrain the heavy LMM.

**3. Resolved Interpretability & Metric Concerns (Reviewer WMmp)**

We addressed specific reviewer questions to clarify our experimental results:
* **Clarification on BLIP Performance:** We clarified that ALA improves BLIP fairness by **32.9%**, significantly outperforming the baseline DeAR. We explained that DeAR fails on BLIP because it modifies the encoder but leaves the decoder's inherent language priors untouched, a limitation ALA specifically resolves.
* **Definition of Fairness (Toxicity):** We justified our "Fairness through Safety" approach (targeting **Non-Toxicity**) over "Equal Toxicity," citing the ethical imperative of safety and the technical absence of visual toxicity ground-truth datasets.

**4. Enhanced Analytical Rigor (Reviewer 1bpB)**

In response to Reviewer 1bpB, we added **Absolute Misclassification Counts** (e.g., Male-to-Female vs. Female-to-Male errors). This granular analysis revealed the baselines' specific directional biases and demonstrated how ALA dynamically adjusts logits to correct these asymmetric errors.

**Conclusion**

With these additional validations covering new models, reasoning tasks, and granular error analysis, we have confirmed that ALA is a robust, model-agnostic, and efficient solution for debiasing multimodal systems. We will integrate these findings into the manuscript to ensure a comprehensive presentation.

Thank you for your time and consideration.

---

### Meta-Review · Area_Chair_Qfu7 · 2025-12-25

**Summary:**

The submission proposes Adaptive Logit Adjustment (ALA), a post-hoc debiasing framework for Multimodal Language Models. By leveraging external classifiers to detect attribute misalignment and utilizing gradient-based importance analysis, the method dynamically adjusts output logits to mitigate stereotypes without requiring model retraining. The reviewers generally appreciated the practicality of this inference-time intervention, though initial concerns were raised regarding the limited selection of evaluated models and the reliance on external classifiers compared to self-steering methods.

**Reviewer Concerns:**

The authors conducted a responsive rebuttal that effectively resolved the major empirical concerns.

Specifically, addressing the critiques from Reviewers WMmP and Jum4 regarding limited scope, the authors successfully demonstrated the method's scalability on the state-of-the-art Qwen2.5-VL-3B model and extended the evaluation to a discriminative "Occupation Recognition" task, proving applicability beyond simple captioning.

While Reviewer 1bpB's theoretical concern regarding the reliance on external classifiers remains inherent to the design, the authors reasonably justified this as a trade-off to avoid the "circular logic" of self-debiasing and to maintain inference efficiency. The addition of absolute misclassification counts further clarified the method's behavior in correcting asymmetric biases.

**Reviewer Scores:**

Given the substantial improvements in the rebuttal, the scores are expected to trend upwards. Reviewers WMmP and 1bpB (Scores: 6) will likely solidify their support as their requests for broader model validation and granular error analysis were met. Reviewer Jum4 (Score: 4), who explicitly cited limited tasks and weak baselines as reasons for the lower score, should be satisfied by the new reasoning task and the strong performance on Qwen2.5-VL, likely moving their recommendation to an Accept.

---

### Decision · Program_Chairs · 2026-01-26

Accept (Poster)